**Brief Communication**

# Clustering the protein universe of life using DIAMOND DeepClust

Benjamin J. Buchfink [1], Émile Barbé [1], Haim Ashkenazy [2], Klaus Reuter [3], John A. Kennedy[3] & Hajk-Georg Drost [1,4] ✉

Relating billions of proteins across the tree of life remains a challenging task for comparative biosphere genomics and artificial intelligence-driven structure prediction. Here we present DIAMOND DeepClust, a cascaded, ultra-fast clustering method enabling planetary-scale organization of protein space, scaling to trillions of sequences while retaining sensitivity at low identity. Aggregating 19 billion biosphere proteins into 544 million nonsingleton clusters, we show that using our DeepClust database, available for download, can enhance structure prediction with AlphaFold2.

As the global biosphere is increasingly sequenced and annotated[1], an unprecedented quality of evolutionary and functional insights can now be harnessed to transform the life sciences. In practice, one such application is the grouping of proteins into related sequence classes that enabled recently celebrated breakthroughs such as protein structure prediction[2], comparative biosphere genomics[3–5] and classification within metagenomic samples[6–9]. These studies are early adopters of leveraging evolutionary information at scale for groundbreaking molecular and functional applications by organizing the entire protein universe for downstream predictive tasks.

Recently, we introduced DIAMOND v.2[10] (a refactored and substantially extended reimplementation of DIAMOND[11]) to meet the user demands for scaling protein search to the tree of life. With DIAMOND v.2, we pledged to support the efforts of the Earth BioGenome project that aims to capture and assemble the genomes of more than 1.8 million eukaryotic species within this decade. In this community quest, we identified the ability to cluster this vast protein sequence diversity space as a key factor currently limiting the association of sequences across large sets of divergent species (Fig. 1 and Extended Data Fig. 1).

Here, we perform a comprehensive experimental study to demonstrate that deep clustering the protein universe of the assembled biosphere exemplified by ~19 billion sequences is already possible today (Extended Data Fig. 2). In the Earth BioGenome era, the ability to reduce the sequence space can notably accelerate protein comparisons when dealing with millions of species and tens of billions of sequences. We estimated that the Earth BioGenome consortium will generate ~27 billion protein sequences when averaging

~15,000 genes per species times ~1.8 million successfully assembled species. Current protein clustering approaches implemented in the standard tools CD-HIT[12], UClust[13] and MMseqs2/Linclust[9] are limited when aiming to cluster billions of proteins with such broad sequence diversity in reasonable time and with sufficient clustering sensitivity at lower identity-boundaries. To overcome this limitation and provide a future-proof software solution, we implemented DIAMOND Deep-Clust, a cascaded clustering method leveraging sensitive protein alignments generated with DIAMOND v.2. Using DIAMOND DeepClust, we reach this clustering milestone to sensitively cluster 19 billion sequences in 18 days on 27 compute nodes (using 250,000 CPU hours in total). This achievement to simultaneously balance speed and sensitivity rather than having to choose between them allows us to substitute MMseqs2/Linclust, UClust, CD-HIT and newly emerging clustering methods such as FLSHclust[14] and HipMCL[4] and meet the user demands of the Earth BioGenome project, where clustering sensitivity across large evolutionary distances is paramount.

As a result of clustering ~19 billion sequences with 30% sequence identity and 90% coverage thresholds, we determined ~1.70 billion clusters with 32% of clusters yielding more than 1 element and 68% denoting singletons (only 1 unique sequence within each singleton cluster). While this majority of singletons suggests the presence of a large pool of putatively new proteins (orphan polypeptides) within the protein universe (Extended Data Fig. 3), these ~1.16 billion unique sequences comprise only ~6% of the full set of 19 billion sequences. The fact that 544 million clusters can capture ~94% of our protein dataset illustrates the potential of deep clustering the protein universe to accelerate protein search across the tree of life.

[1]Computational Biology Group, Max Planck Institute for Biology Tübingen, Tübingen, Germany. [2]Department of Molecular Biology, Max Planck Institute for Biology Tübingen, Tübingen, Germany. [3]Max Planck Computing and Data Facility, Garching, Germany. [4]Digital Biology Group, Faculty of Life Sciences, University of Dundee, Dundee, UK. ✉e-mail: hdrost001@dundee.ac.uk

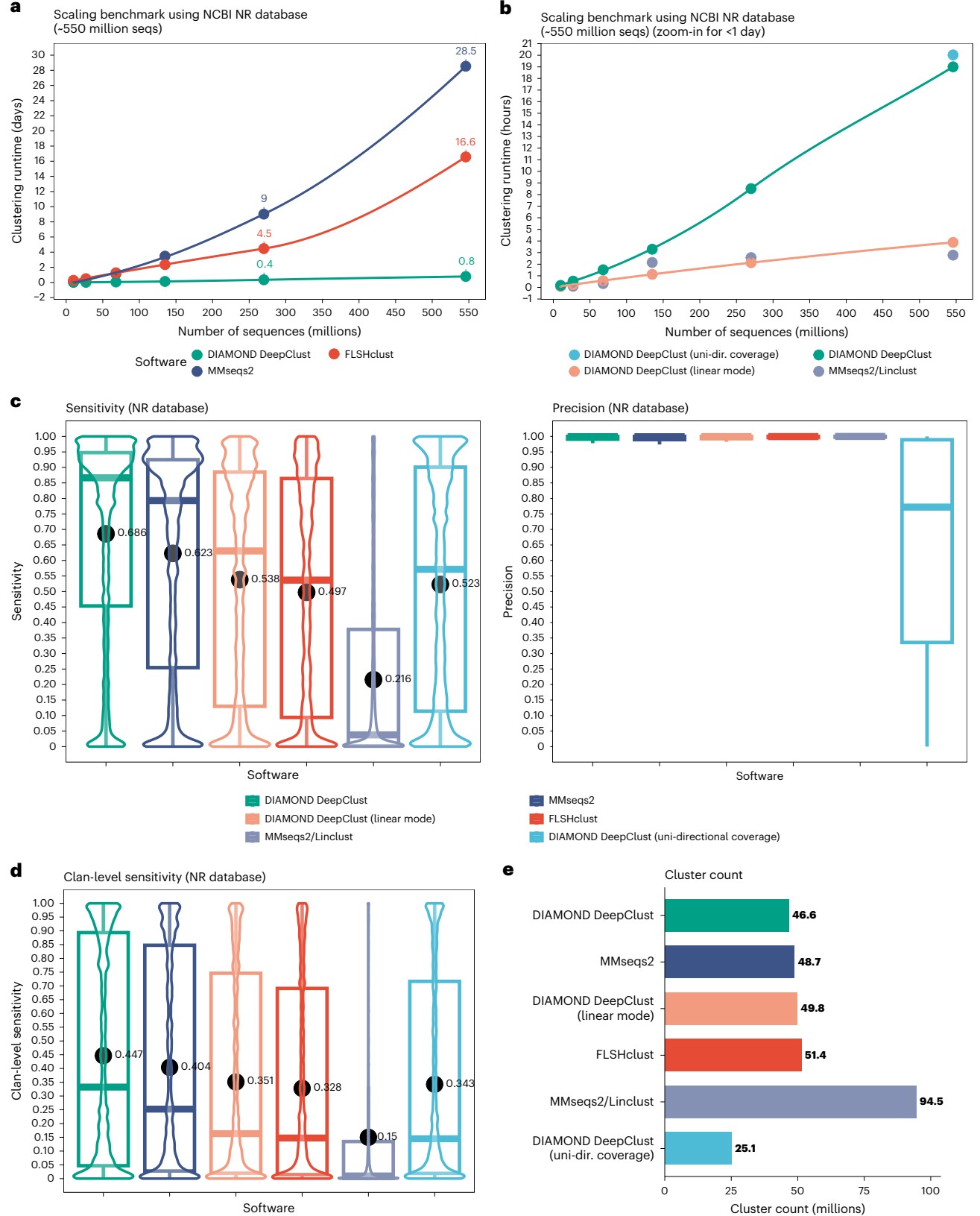

**Fig. 1 | Benchmark of the clustering performance of DIAMOND DeepClust, MMseqs2 and FLSHclust.** Computational benchmarks are shown for deep clustering the NCBI NR database currently storing ~546 million protein sequences (seqs) using bi-directional coverage (unless indicated otherwise) and no sequence identity threshold. **a**, Clustering runtimes for clustering the NR database and subsamples of increasing size on a 64-core server are shown in days. **b**, Same as **a** but only showing tools with a runtime less than 1 day in hours.

**c**, The resulting distribution of sensitivity and precision over the 150 million annotated input sequences with respect to Pfam domain architectures of compressing the NCBI NR database. **d**, The sensitivity when considering different Pfam families in the same clan as equivalent. **e**, The number of clusters produced by the tools (the line inside each box is the median; box edges show the 25th (Q1) and 75th (Q3) percentiles; whiskers extend to the most extreme values within 1.5× IQR of the quartiles).

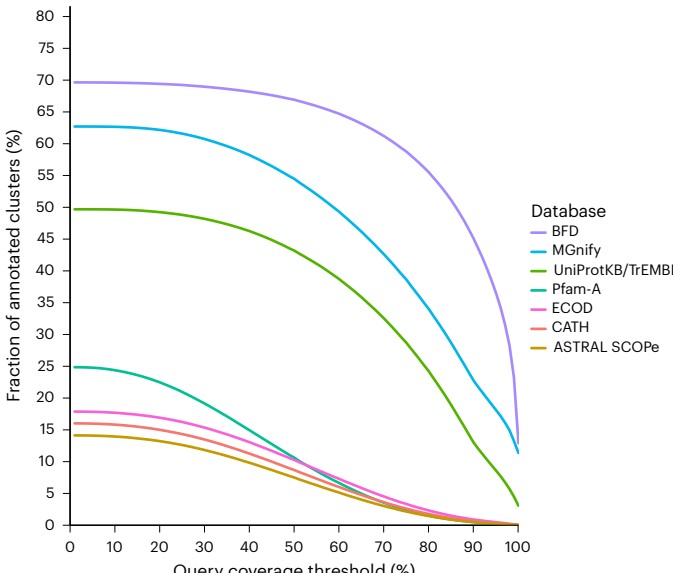

**Fig. 2 | Characterization of the protein universe clustered by DIAMOND DeepClust with respect to existing databases.** Shown is the fraction of representatives from clusters of size ≥3 of our 19 billion sequences dataset that can be annotated based on the given databases depending on the sequence coverage threshold of the representative (sample size 1,000,000 representatives).

For our dataset, we found that 92% (17.8 billion sequences) can be compressed into 335 million representative sequences for downstream analyses (Extended Data Fig. 2).

Notably, recent breakthroughs in protein structure prediction such as AlphaFold2 were largely derived from the use of the Big Fantastic Database (BFD)[2], a public collection of diverse protein sequences containing 345 million clusters and 61 million clusters with at least 3 members. While currently holding the status of the largest collection of deep-clustered protein sequences, the result of our experimental study yielding 335 million clusters with at least 3 members represents a 5.5-fold increase in sequence diversity (as measured by the cluster count) compared with the BFD. We estimated that 118 million protein families from our DeepClust database are new, because they could not be mapped to the BFD using HHblits[15] with at least 60% query sequence coverage (Fig. 2). We performed deep clustering using the same thresholds used to construct the BFD so that cluster count results from our 5.5-times larger DeepClust database are directly comparable with the results obtained from the BFD. We also show that integrating our dataset with AlphaFold2 has substantial potential to improve structure prediction for sequences not sufficiently represented in smaller databases (Extended Data Fig. 4). The same conclusion was reached by DeepMSA2[16] (and personal communication with the DeepMSA2 team), which searches through up to 40 billion protein sequences using HMMER to support its structure predictions. As such a search through an un-clustered database of this size may require 90–400 CPU hours for a single query (Supplementary Notes), deep clustering is essential for making such computations feasible at scale.

Previous methodologies to cluster protein sequences such as CD-HIT and UClust were not designed to scale to millions of species and perform suboptimally when attempting to cluster datasets deeper than 50% sequence identity at that scale. Although MMseqs2[9] presented a considerable advancement over CD-HIT and UClust, it still suffers from comparatively low performance when clustering at high alignment sensitivity, thereby introducing an analytics bottleneck when attempting to scale to >27 billion estimated Earth BioGenome sequences covering the full breadth of biospheric protein space.

The newly emerging method FLSHclust's[14] comparatively better scaling behavior to billions of proteins comes at the expense of substantially reduced sensitivity due to a more greedy approach that foregoes rigorous all-versus-all alignment.

To benchmark DIAMOND DeepClust against MMseqs2 and FLSHclust, we clustered the National Center for Biotechnology (NCBI) nonredundant (NR) database containing ~546 million sequences using a bi-directional coverage criterion without a sequence identity threshold (deep clustering) (Fig. 1). DIAMOND DeepClust solved this problem for deep clustering in 19.0 on a single server equipped with 64 cores compared to 29 days using MMseqs2, running 36-fold faster. DIAMOND DeepClust exhibited similar clustering sensitivity (68.6% versus 62.3%) as MMseqs2 at comparable precision (95.5% versus 95.5%), both measured with respect to annotated Pfam domain architectures. DIAMOND DeepClust was 21-fold faster than FLSHclust (which ran for 17 days) while outperforming its sensitivity (68.6% versus 49.7%) at comparable precision (95.5% versus 96.5%). For comparison purposes, we also show a clustering run using DIAMOND DeepClust with a uni-directional coverage criterion. The resulting clustering was inferior in both sensitivity (52.3%) and precision (64.9%), showing the advantage of bi-directional coverage clustering when aiming to build clusters of conserved protein families. In addition, we note that our method strongly outperforms a recently published HipMCL-based pipeline[4] that reportedly required 570,000 CPU hours to cluster 1.2 billion protein sequences compared to DIAMOND DeepClust processing 550 million sequences in 1,200 CPU hours, while the HipMCL pipeline is also not adapted for deep clustering due to the 70% sequence identity threshold that was used.

While sensitive all-versus-all alignment will remain the standard for deep clustering, methods with linear scaling behavior are a powerful alternative to handle large datasets highly efficiently. The linear mode of DIAMOND DeepClust ran for 3.9 hours or 102-fold faster than FLSHclust at comparable sensitivity (53.8% versus 49.7%) and precision (95.9% versus 96.5%). While MMseqs2/Linclust was the fastest tool running for 2.8 h, its sensitivity of 21.6% was far behind the other tools (at a precision of 97.5%). We also note that MMseqs2/Linclust is limited to processing 4 billion sequences and running on a single compute server, while we engineered the linear mode of DIAMOND DeepClust to fully leverage massively parallel computations on compute clusters or in the cloud scaling to trillions of sequences and petabytes of data (Methods) (Extended Data Fig. 1).

## Online content

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

# Methods

## Algorithmic overview of DIAMOND DeepClust

**Representative-based clustering.** Following an analogous strategy as the gold standard approaches implemented in CD-HIT[12] and UCLUST[13], we define a clustering of an input dataset of protein sequences as a subset of representative sequences such that any input sequence lies within a user-defined distance threshold of at least one representative sequence. As a result, each input sequence will be assigned to one representative sequence. This threshold setting (sequence alignment identity, coverage and $e$ value) is also referred to as the clustering criterion. For the purpose of this study, in addition to a basic $e$ value threshold of 0.001, we require that a pairwise local alignment between sequences satisfy a specific minimum sequence identity and length coverage of both the representative and the coclustered (nonrepresentative) sequences (bi-directional coverage clustering). In addition, we also carried out clustering runs using uni-directional coverage clustering that only requires coverage of the co-clustered (nonrepresentative) sequence. In this context, sequence coverage refers to the length of the range spanned by a local alignment divided by the length of the respective sequence.

**Greedy vertex cover.** Starting from a set of alignments that meet the user-specified or default clustering criterion, we compute a set of representative sequences by first encoding the alignments as a directed graph G where nodes represent individual protein sequences and edges denote pairwise local alignments between them, whereby a directed edge from sequence A to sequence B indicates that A can represent B according to the clustering criterion. In a second step, we apply the greedy vertex cover algorithm[17] on the alignment-graph G to determine a near-minimal covering set of graph vertices. The algorithm repeatedly selects the vertex with the highest node outdegree and removes it from the graph along with its out-neighbors to form a new cluster until the graph is completely clustered. The greedy vertex cover approach can be generalized to also include indirect neighbors of the designated representative sequence up to a certain search depth on the graph into the newly formed cluster. Here an infinite search depth would mean to form clusters that correspond to the connected components of the graph. DIAMOND DeepClust uses a search depth of one for the first two rounds of cascaded clustering, and a depth of three for the remaining rounds. While the greedy vertex cover algorithm is useful to find a minimum covering set of representatives and minimize the cluster count, it does not attempt to identify weak edges in the graph between densely connected communities that could indicate false positives and should be ignored. We plan to integrate Markov Clustering[18] into DIAMOND DeepClust to address this limitation.

**Minimizer sampling.** In the first round of cascaded clustering, we subsample the seed space using minimizers[19] with a window size of 12 that we empirically found to provide a good balance between speed and sensitivity. While similar to the sketching approach proposed by MMseqs2/Linclust[9], our approach samples a number of minimizers that depends on the length of the sequence instead of a fixed number per sequence.

**Linear-stage clustering.** MMseqs2/Linclust proposed to achieve linear computational scaling of comparisons by considering only seed hits against the longest sequence for identical seeds rather than trialing all possible combinations[9]. This heuristic is sufficient in the context of uni-directional coverage clustering to find meaningful representatives as the seeds at this stage are selected to be highly specific and the longest sequence is a priori the most likely to maximize recruitment of member sequences due to the uni-directional coverage criterion. It is insufficient for bi-directional coverage clustering however, as it considers comparisons that cannot possibly satisfy the bi-directional coverage criterion based solely on the length of the sequences. We observe that for a coverage threshold $c$ and a length ratio $\frac{l_1}{l_2}$ ($l_1 \le l_2$) between two sequences, a local alignment of the two sequences can only reach a coverage of $c$ of both sequences if $\frac{l_1}{l_2} > c - \varepsilon$ holds (with $\varepsilon > 0$ being some tolerance to account for gaps). To make use of this property, our algorithm will conduct comparisons for a group of $n$ sequences $s_1, \cdots, s_n$ of lengths $l_1, \cdots, l_n$ ($l_i \ge l_{i+1}$) containing the same seed as follows. An initial range of $[i;j]$ is constructed with $i = 1$ and $j = \max\{k\epsilon[i;n] : \frac{l_k}{l_i} \ge c + \delta\}$. All sequences with index in the range $[i;j]$ are then compared to the 'median' sequence of index $\lfloor \frac{i+j}{2} \rfloor$. If $j = n$ the algorithm aborts, otherwise the interval is then advanced by setting $i = \min\{k\epsilon[i+1;n] : \frac{l_k}{l_{j+1}} \ge c + \delta\}$ and $j$ as above, followed again by comparison against the median sequence and advance of the interval. We note that instead of subtracting a tolerance $\varepsilon$ from the coverage threshold, we add a positive value $\delta$ ($\delta = 0.05$ by default) to make it more stringent, which works better in practice as it will increase the number of comparisons. This algorithm is designed to select a subset of potential representative sequences likely to maximize recruitment of cluster members in the context of a bi-directional coverage criterion.

**Cascaded clustering.** As exhaustive all-versus-all alignment of protein datasets consisting of hundreds of millions of sequences is prohibitively expensive, we approach this issue by adopting cascaded clustering (personal communication with the DeepMSA2 team) to gradually construct larger sequence clusters in several rounds of comparison with increasing alignment sensitivity (iterating between modes: –fast, default, –sensitive, –very-sensitive, –ultra-sensitive, corresponding to the sensitivity modes of our DIAMOND v.2 aligner[10]). A round of cascaded clustering performs self-comparison of the input sequences at a round-specific sensitivity. We then compute representative sequences from the resulting alignments as explained above under greedy vertex cover clustering, which are then passed on to the next round of cascaded clustering and subjected to an all-versus-all search at increased sensitivity. Depending on the desired clustering depth, two to six of these alignment rounds are chained until reaching sufficient sensitivity such that most representatives within clustering distance have been discovered. The first round of cascaded clustering uses the linear-stage algorithm in conjunction with minimizer subsampling. Contrary to MMseqs2 that uses a single linear round, we add a second linear round without minimizer subsampling, which we found to provide an additional 36% reduction of the sequence space (in case of clustering the NR database) without requiring expensive all-versus-all comparison. In addition, we found it advantageous to use a more stringent clustering criterion in earlier rounds of cascaded clustering and relax it gradually in later rounds. More specifically, we add +7% to the coverage and sequence identity thresholds up to the round using default mode sensitivity, and +5% to the coverage threshold in the round using the '–sensitive' mode. For example, when the base identity cutoff is set to 20% then all individual rounds up to the default mode will use a 27% threshold. The algorithm is memory efficient (Supplementary Discussion 1) and permits adding new sequences to an existing clustering (Supplementary Discussion 2).

**Self-alignment optimization.** We optimized self-alignment of the representative databases by taking advantage of the symmetry of queries and targets in the seeding stage, avoiding the evaluation of redundant seed hits, and thus doubling the performance of this computation. This is accordingly considered when the database is processed in blocks by eliminating redundant block combinations.

**Bi-directional coverage optimization.** As explained above, many sequence pairs can be excluded from satisfying a bi-directional coverage criterion based on the ratio of their respective lengths. We take full advantage of this property in the earliest stage of the pipeline for all-versus-all alignment computations. In the context of the double-indexing algorithm of DIAMOND, this is accomplished by

sorting the seed index tables by the length of the sequences in descending order. Processing the hits of a seed, for a given query sequence of length $l_q$ and a set of $n$ target sequences of lengths $l_1, \cdots, l_n$ we determine a range $[i;j]$ of target sequences such that $i = \min\{k \epsilon[1;n] : \frac{l_k}{l_q} \geq c - \varepsilon\}$ and $j = \max\{k \epsilon[1;n] : \frac{l_k}{l_q} \geq c - \varepsilon\}$ (using a value of $\varepsilon = 0.05$ by default). We then compare the query sequence to all target sequences within the index range $[i;j]$. To process the next query sequence, both indexes $i$ and $j$ are then incremented until the conditions above are met again. This is only a minimal computation due to the length sorting of both the query and the target index table.

**Multiple spaced seeds for linear mode of DIAMOND DeepClust.** The Linclust approach[9] as published depends on choosing long and specific seeds that in turn limits its sensitivity and applicability to deep clustering. To overcome this limitation, we use multiple spaced seeds[20] to retain specificity while increasing the sensitivity of the seeding. Previous approaches to compute multiple spaced seeds[21,22] used theoretical models to compute a near-optimal set of multiple spaced seed patterns. We propose to empirically learn these patterns based on a collection of real protein alignments. To this end, we aligned a sample of 10,000 sequences from the UniRef50[23] database against the whole database using DIAMOND v.2 in very-sensitive mode, retaining at most 300 alignments per query, yielding a collection of 1.47 million alignments. Our algorithm parses the collection of alignments and determines the seed shape pattern of a defined weight and maximum length that would hit the largest collection of alignments. This pattern is then saved and all alignments that it hits are removed from the collection, at which point the algorithm repeats until a predefined number of seed shape patterns has been designed. When running in linear mode, DIAMOND DeepClust uses cascaded clustering with the first two rounds being identical to the workflow described in that chapter. The third round also uses the linear-stage algorithm in conjunction with 30 seed shapes with a weight of 10 and a search depth of 2 for connected component clustering.

**Multi-node parallelization.** Clustering on the protein level presents an important component of addressing the challenges of learning from exponentially growing experimental data, with current databases containing several tens of billions of sequences. While MMseqs2/Linclust marked an important advancement in providing a sequence clustering algorithm that scaled linear with dataset size, for practical purposes its scalability to large datasets is impeded by several factors. First, it can only process datasets of up to 4 billion sequences due to the use of 32-bit sequence identifiers. Second, their software demands that the entire input database can be efficiently accessed in a random-access manner. This means that the entire database either needs to fit into memory on the host system or that the storage system needs to provide such efficient access, an assumption that usually only holds true if the data resides on a fast local solid-state drive but typically fails on cluster or cloud-based storage systems. Finally, it is unable to parallelize across multiple compute nodes, running a clustering job for a single dataset on only a single compute server (we note that while the regular `mmseqs cluster` workflow based on all-versus-all alignment can run on multiple nodes using MPI, the `Linclust` part of the workflow only runs on a single node). In our implementation of the linear mode of DIAMOND DeepClust, we aimed to address these limitations, providing a tool capable of massively parallel cluster or cloud-based computations and scaling to trillions of sequences and petabytes of data. While clustering based on all-versus-all alignment is trivially parallelizable by splitting the input database into chunks and computing an all-versus-all alignment of these chunks, the Linclust algorithm cannot be parallelized by just partitioning the input database due to the global dependency of comparing each sequence to the longest sequences with which it shares a common seed.

Our algorithm is implemented as a series of modular computing steps that operate on tables that can span across many storage volumes on a distributed file system, ingested and processed in parallel on many compute nodes. The first step is reading the input database and producing an output table of seed, sequence ID and sequence length triplets according to the seeding strategy used by the clustering. Next, this table is sorted on the seed column using our own parallel radix sort implementation, optimized toward distributed external sorting of large datasets ranging up to petabytes in size. We perform recursive radix clustering on the highest relevant bits of the key that were not yet subjected to sorting until the size of a bucket is small enough to be processed in-memory on a single compute node. The now sorted seed table is then processed seed-by-seed. A pair table containing pairs of putative representative and cluster member sequences is produced by assigning each sequence to the longest sequence in the same seed groups in case of uni-directional clustering, or by applying our modified logic in case of bi-directional clustering (section 'Linear-stage clustering'). The pair table is then radix sorted using our implementation on the representative sequence and read linearly, building a table assigning database sequences to chunks of sequences that need to be processed together, with the chunks limited in size to what can be processed in-memory on a single compute node. Next, the output table from this previous step is then radix sorted using our implementation on the sequence ID and processed linearly together with the input database, storing each sequence into the chunks that it has been assigned to. These chunks are then processed independently one-by-one, computing all associated alignments and storing the results as a table of directed graph edges where an edge from sequence A to sequence B means that A can represent B according to the clustering criterion. The last two steps can optionally be processed in multiple rounds to save storage space. Finally, we reduce the inbound edges of each node to the edge whose source is the longest of the possible representative sequences in case of uni-directional coverage clustering, or which has the highest outdegree in case of bi-directional coverage, and output the transitive closure of the resulting graph as the final clustering. All tables are partitioned into 256 radix buckets on creation to reduce the cost of radix sorting.

To benchmark our implementation, we ran the linear mode of DIAMOND DeepClust on our experimental study dataset of 22 billion sequences, clustering at 90% sequence identity and 80% uni-directional coverage, performing several runs on 1, 2, 4, 8, 16 and 32 compute nodes (equipped with dual AMD EPYC Genoa 9554 CPUs with 128 cores and 512 GB of RAM). We observe that our implementation exhibits nearly ideal scaling behavior to multiple nodes, reducing the runtime from 15.3 h on a single node to 35 min on 32 nodes (Extended Data Fig. 1).

The multi-node feature is available since DIAMOND v.2.1.14 and can be accessed by running the `DIAMOND Linclust` workflow and setting the `-parallel-tmpdir` parameter to a shared working directory on all nodes. It supports bi-directional and uni-directional coverage clustering at any sequence identity threshold including deep clustering using multiple spaced seeds.

**Gapped alignment computation.** We produced a new vectorized Smith Waterman implementation based on a modified SWIPE[24] approach that was originally developed for the first DIAMOND version[11], but dropped out of its code base soon after the initial release. While SWIPE vectorized the Smith Waterman algorithm by computing alignments of the same query against multiple targets, we generalize this approach to computing alignments of multiple independent query–target pairs. This is accomplished by using score profiles for the queries that store alignment scores along the sequence for each of the amino acid residues. For computing one column of the dynamic programming matrix, we maintain pointers into these profiles for each SWIPE lane and apply an AVX2-optimized matrix transposition to interleave the query–target scores for each dynamic programming cell into the

same register, then compute the cell updates according to the standard SWIPE logic. Contrary to the original SWIPE design, this approach permits the computation of banded and anchored alignments, which in turn also enable optimizations such as the cheap determination of alignment start and end coordinates as well as X-drop termination. Compared to our implementation of the original SWIPE algorithm, we measured ~20% higher cost per cell update for this approach on the Intel Ice Lake architecture, while the number of cell updates can be greatly reduced as discussed above.

## Benchmarks

**Benchmark design.** Our clustering benchmark is based on the NCBI NR database[25] downloaded in June 2023, containing 545,815,195 sequences. As we aimed to benchmark both the precision and the sensitivity of a clustering based on an annotated ground truth, we aligned all sequences against Pfam-A 33.1[26] and determined[27] domain architectures for all sequences with 80% residue-level annotation with Pfam families, resulting in a dataset of 149,824,975 annotated sequences (Supplementary Notes), which we use for the evaluation. To evaluate a clustering run, we define the sequence-level sensitivity as the number of sequences in the same cluster of the same domain architecture, divided by the number of sequences of this architecture in the input database. We define the sequence-level precision as the number of sequences in the same cluster of the same domain architecture, divided by the size of the cluster. For the purpose of determining precision, we consider annotations with different Pfam families belonging to the same Pfam clan as equivalent, as members of the same Pfam clan are assumed to possess a remote homology. For each clustering run, we visualize the distribution of the sequence-level sensitivity and precision (Fig. 1c) and average these numbers using the arithmetic mean for a compounded value.

We conducted clustering runs for DIAMOND DeepClust, MMseqs2[9] and FLSHclust[14] on a 64-core server with 2 TB of RAM using the full NR database as input. To determine sensitivity and precision, non-annotated sequences in the output are ignored. We opted for this benchmark design instead of just clustering the subset of annotated sequences as the focus of this work is computational performance and scalability. We highlight that since the tools have no way of knowing which of the input sequences are annotated, this benchmark design does not provide any kind of unfair advantage to our own tool.

As a single clustering run of MMseqs2 takes more than 3 weeks for the NR database, we did not find it practical to repeat this computation many times with different parameter settings to produce a receiver-operating characteristic curve-like figure. Instead, we attempted to determine the best possible parameter settings of MMseqs2 (maximizing the sensitivity) while providing at least 95% precision (Supplementary Notes) and used these settings for a single clustering run of the NR database. We chose the parameters of DIAMOND DeepClust accordingly to provide a comparable result to MMseqs2 and conducted a single run with these parameters. We ran FLSHclust using an 80% coverage threshold and a 20% sequence identity threshold as a three-step cascaded clustering run (Supplementary Notes).

For all tools, we conducted runs with identical parameters on randomly downsampled databases of the NCBI NR, using sample sizes of 270 million, 135 million, 68 million, 27 million and 10 million sequences (Fig. 1).

In addition to the runs presented in (Fig. 1), we also conducted a run of DIAMOND DeepClust using a 30% sequence identity threshold and a 90% uni-directional coverage threshold (corresponding to the settings used for the experimental study). Compared to the run using no sequence identity threshold and an 80% uni-directional coverage threshold, this change reduced the sensitivity from 52.3% to 42.5% and increased the precision from 64.9% to 72.3%. The runtime on the benchmark system was 39 hours and 40 min.

**Limitations of DIAMOND DeepClust and future improvements.** While DIAMOND DeepClust allows users to efficiently scale protein sequence clustering to billions of sequences, our method has its limitations. In particular, the maximum sensitivity achievable on the level of pairwise sequence comparisons is a limiting factor in detecting very distant homologs. The aim of this software and accompanying experimental study is to provide a scalable software solution and clustering methodology to associate the billions of protein sequences available today through local pairwise alignment. In future work, we will then focus on refining their relationships in terms of sequence alignment composition and sensitivity beyond pairwise alignments. For future developments of DIAMOND DeepClust we therefore predict that integration of profile-based searches will further increase its sensitivity.

While our software aims to find sequences with homologous origin, the difference between an ortholog and a paralog is an important distinction in the context of phylogenomics. We believe that further work is needed to enable this analysis for when millions of genomes will be available.

## Experimental study

Recently, we introduced DIAMOND v.2[10] to unlock the familiar functionality of a BLASTP search for tree-of-life-scale applications in the Earth BioGenome era[28–31]. Searching protein sequences against millions of species and yielding trillions of pairwise alignments still remains a computational challenge. Overcoming this bottleneck requires extensive dimensionality reduction of protein sequence space into sequence clusters to perform pairwise alignments only on the set of representative sequences rather than all sequences.

Using DIAMOND DeepClust, we performed an experimental study to showcase the power of dimensionality reduction through sequence clustering when the Earth BioGenome project will have successfully sequenced and assembled all ~1.8 million species. For this purpose, we collected ~22 billion protein sequences across all kingdoms of life to match the order of magnitude and sequence diversity space of the estimated ~27 billion eukaryotic protein sequences planned to be generated as part of the Earth BioGenome project (assuming ~1.8 million species times an average of ~15,000 genes per species). We note that compared to our collection of ~22 billion sequences (~19.4 billion deduplicated sequences), the Earth BioGenome set will include a much broader sequence diversity space derived from eukaryotes (while our current dataset is enriched in proteins mostly derived from microbial metagenomic samples). We projected that our software will scale to the full Earth BioGenome dataset and yield <3.99 billion (upper bound estimate) clusters when deep clustering at 80% bi-directional coverage, feasible on a single server using the linear mode (Supplementary Discussion 3).

As a result of clustering ~19 billion deduplicated sequences with 30% sequence identity and 90% uni-directional coverage thresholds across the tree of life, we determined ~1.70 billion clusters with 32% of clusters yielding more than 1 element, 12% of clusters holding more than 5 elements and 68% of clusters with only 1 element (singletons) (Extended Data Fig. 2). We chose this clustering criterion equivalent to the criterion used to create the BFD to make the result comparable in terms of cluster count, even though our main benchmark is focused on bi-directional clustering. This result shows that while 68% of clusters contain unique sequences, these ~1.16 billion singletons represent only 6% of the 19 billion sequences defining our protein universe, which begs the question whether these distinct proteins are derived from new orphan genes or whether they represent assembly and annotation artifacts (Extended Data Fig. 3 and Supplementary Discussion 4).

To quantify the relatedness of our clusters to existing protein databases, for a sample of 1 million representative sequences from the clusters containing 3 or more sequences, we aligned them against UniProtKB/TrEMBL[32] Release March 2025 and MGnify[33] Release April 2024 using DIAMOND v.2 in ultra-sensitive mode, against Pfam-A

release v.37.4[26] (June 2025) using HMMER[27] and against CATH[34] v.4.3.0, ECOD[35] release 285, ASTRAL SCOPe[36] v.2.08 and the BFD[2] using HHblits[37] (*e* value threshold 0.001). For each query representative sequence, we determined its coverage maximizing over the hits to the target database. We accumulated query coverage over all hits between a query–target pair and over all hits to the target database for the domain-level databases. We show the fraction of representative sequences that can be annotated for a given target database depending on the query coverage threshold (Fig. 2). For example, 64.7% of our representatives are covered over at least 60% of their range by a BFD sequence. Projected to our whole dataset of 335 million clusters of size of at least 3, this would correspond to 118 million new sequence clusters not represented in the BFD.

**Protein structure prediction.** Protein sequence clustering is an important upstream application to protein structure prediction as realized by current state-of-the-art methods such as AlphaFold2[2,38,39]. The predictive power is drawn from the evolutionary information stored in larger clustered databases, to this end AlphaFold2 uses both the BFD (containing 2.4 billion sequences) and the MGnify[33] database (containing 1.2 billion sequences). Here we investigate the potential of using even larger protein databases to improve structure prediction based on our experimental study database created by clustering 19 billion sequences. To find candidates likely to benefit most from using our larger database, we randomly sampled 10,000 clusters from the clusters containing between 30 and 100,000 sequences. We chose a minimum size of 30 as this threshold is reportedly significant for the prediction accuracy of AlphaFold2[2]. We aligned the representative sequences of these clusters against the BFD using HHblits[37] and against MGnify[33] and UniRef30[32,40] using JackHMMER[41]. We selected the 732 sequences that in total had less than 30 hits against these databases and used them as input for structure prediction running AlphaFold2. Out of these, we selected the 473 sequences that had an AlphaFold2 multiple sequence alignment (MSA) depth of less than 30 and ran AlphaFold2 again, this time modified to use our experimental study database for homology search and MSA construction instead of BFD and MGnify. We report AlphaFold2-generated pLDDT (predicted local distance difference test) scores for these predictions depending on the database used, which is AlphaFold2's internal confidence measure shown to correspond well with experimental verification (Extended Data Fig. 4). Using the DIAMOND DeepClust database, the pLDDT score increased for 366 sequences and decreased for 107, and improved on average by 7.73 from 52.9 to 62.6. These results indicate a substantial potential to improve AlphaFold2 structure predictions for sequences not well represented in smaller databases by leveraging protein clustering at the scale of the known protein universe. We point out that while our improved structure prediction is nonrandom (Wilcoxon rank sum test $P < 2.2 \times 10^{-16}$), due to the high computational cost (this analysis needed around 100,000 CPU hours) of fold prediction and associated computations, our sample size was limited to 473 proteins. Further studies need to focus on the scalability of our finding to the entire protein universe. Note that we also provide code for using our database in ColabFold[38] ('Code availability').

**Clustering.** A total of 22,788,215,153 publicly accessible protein sequences were retrieved from JGI IMG[42], SRC and MERC[43], MGnify[33], Metaclust[42], NCBI NR[25], AGNOSTOS[44], MetaEuk[45], SMAGs[46], TOPAZ[47], GPD[48], NovelFams[49] and MGV[8] databases during March to April 2022. After preprocessing and deduplication (Supplementary Notes), we clustered the combined input file using a cascaded clustering approach in four rounds at increasing sensitivity employing the DIAMOND modes –faster, –fast, default and –sensitive, also using the option to linearize the comparison in the first round as described in the cascaded clustering section. The clustering criterion was 90% uni-directional coverage of the cluster member sequence and 30% approximate sequence

identity, corresponding to the parameters used to generate the Alpha-Fold2 BFD[2]. The sensitivity level of DIAMOND v.2 also roughly corresponds to the BFD creation. The clustering computation of the input sequences ran for 6.38 days on a single high-memory 72-core node for the first round, 36.8 h on 16 nodes for the second round, 20.1 h on 16 nodes for the third round and 9.63 days on up to 27 nodes for the fourth round. In total, the computation consumed ~255,000 CPU hours.

We make the results of our clustering computation publicly available. Download links for a BLAST database and FASTA files containing the cluster representative sequences, a TSV file mapping the accession of each input sequence to its representative sequence, a BLAST database and Parquet file of all cluster member sequences, an index and scripts to quickly extract individual clusters are provided ('Code availability' and 'Data availability').

### Reporting summary

Further information on research design is available in the Nature Portfolio Reporting Summary linked to this article.

### Data availability

The results of clustering 19 billion protein sequences as part of our experimental study is available at https://objectstore.hpccloud.mpcdf.mpg.de/deepclust/index.html, including the BLAST database and FASTA file of representatives, TSV file mapping cluster members to representatives, BLAST database and Parquet files containing all cluster member sequences, index files for fast retrieval and ColabFold database. Source data are provided with this paper.

### Code availability

DIAMOND DeepClust is available as Open Source Software under the GPL3 license via GitHub at https://github.com/bbuchfink/diamond. The scripts needed to run the benchmarks and analyses and produce the figures are available via GitHub at https://github.com/drostlab/deepclust-data. Code for using the experimental study database with ColabFold is available via GitHub at https://github.com/drostlab/deepclust_colabfold. Code for extracting sequences from Parquet files is available via GitHub at https://github.com/drostlab/deepclust_dataretrieval.

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

## Acknowledgements

We thank D. Weigel for his generous sponsorship, discussion and overall support of this work. We furthermore thank the members of the Drost and Weigel laboratories for extensive discussions during laboratory meetings and the Max Planck Computing and Data Facility for providing computational resources. We also thank J. Katz for dedicating her Bachelor Thesis to protein clustering and for valuable discussions in the early phases of this project. We express our deep gratitude to A. Tomescu, V. Alva, A. Gynter, F. Dias, A. Grigorjew, A. Dallaire, K. Ouyang, V. Fernández Roces, J. S. Lotharukpong and L. Maischak for trialing early versions of DeepClust in the context of their work and for providing valuable feedback and raising constructive discussions. We thank H. Altae-Tran for support with the FLSHclust software, R. Edgar for providing a free 64-bit license of USEARCH and W. Zheng, P. L. Freddolino and Y. Zhang for providing DeepMSA2 benchmarking data. H.-G.D. is also supported by a Royal Society Wolfson Fellowship (grant no. RSWF\R1\241004). We thank the BMBF-funded de.NBI Cloud within the German Network for Bioinformatics Infrastructure (de.NBI) (grant nos. 031A532B, 031A533A, 031A533B, 031A534A, 031A535A, 031A537A, 031A537B, 031A537C, 031A537D, 031A538A) for computational support (awarded to H.-G.D.). This work was supported by the Max Planck Society.

## Author contributions

H.-G.D. conceived and led the study. H.-G.D. and B.J.B. designed the experimental study, B.J.B. designed and implemented the algorithms and performed the benchmarks, H-G.D., B.J.B., H.A., E.B., K.R. and J.A.K. performed the experimental study. E.B. performed the AlphaFold2 analysis. B.J.B., H.-G.D., H.A., E.B., K.R. and J.A.K. analyzed and interpreted the results. B.J.B. and H.-G.D. wrote the paper with contributions from H.A. and E.B. All authors have read and approved the final version of the paper.

## Funding

## Competing interests

The authors declare no competing interests.

## Additional information

**Extended data** is available for this paper at https://doi.org/10.1038/s41592-026-03030-z.

**Correspondence and requests for materials** should be addressed to Hajk-Georg Drost.

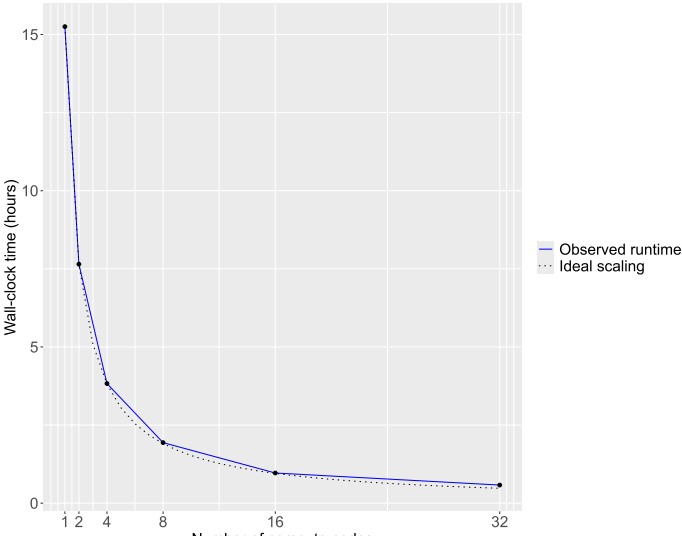

**Extended Data Fig. 1 | Multi-node scaling of DIAMOND DeepClust (linear mode).** Shown is the wall-clock runtime of clustering the 22 billion protein sequences from our experimental study at a 90% identity threshold on the given number of compute nodes using the linear mode of DIAMOND DeepClust and the expected runtime in case of ideal scaling, obtained by dividing the runtime on a single node by the number of nodes. As observed, our multi-node implementation achieves nearly ideal scaling behavior.

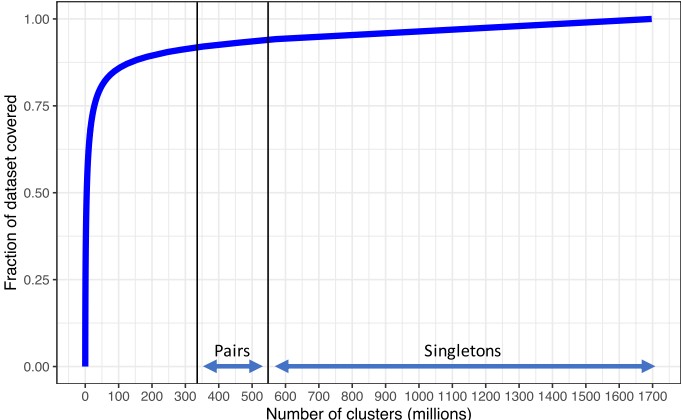

**Extended Data Fig. 2 | Cluster representation for the experimental study.**
Shown is the number of clusters generated with DIAMOND DeepClust against the fraction of the 19.4 billion input proteins that are covered by the clustering. The vertical lines indicate the start of clusters of size two and one (from left to right). The result illustrates that ~335 million representatives can capture 92% (~17.48 billion sequences) of the protein universe exemplified by 19 billion sequences.

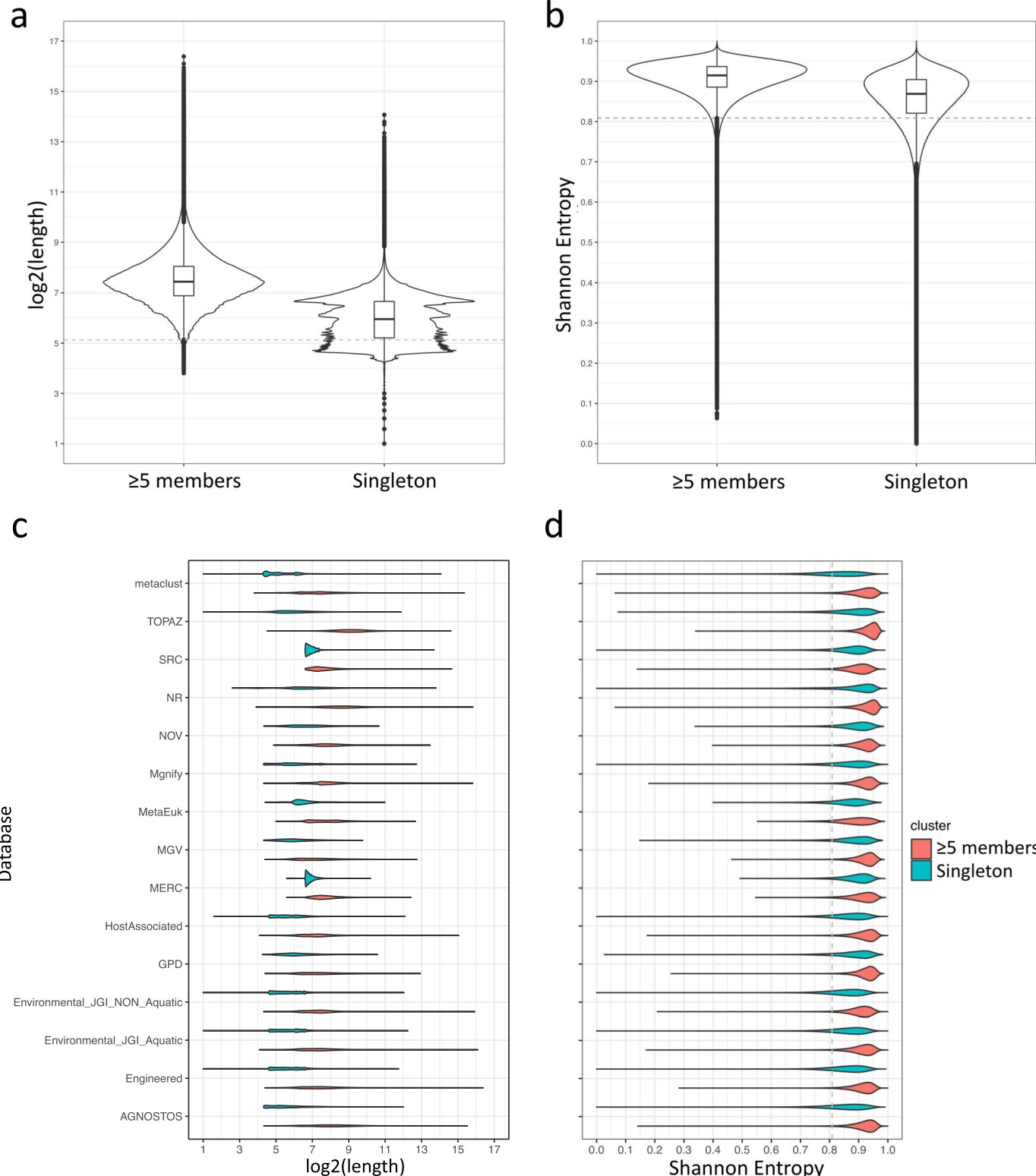

**Extended Data Fig. 3 | Characterization of the 1.7 billion clusters generated by DIAMOND DeepClust.** Illustrated are sequence length and Shannon entropy for cluster representatives. The length and Shannon entropy (sequence randomness) were computed for the representatives of clusters with at least five members (204 million) compared with the singletons (1.16 billion). IQR outlier cutoff was determined based on the distribution computed for clusters with at least five members, denoted as dashed lines. **a.** Sequence length distribution of non-singleton versus singleton clusters. Note that some databases allow 1AA

as the smallest unit defining a protein, **b.** Shannon entropy distribution of non-singletons versus singleton clusters. (the line inside each box is the median; box edges show the 25th (Q1) and 75th (Q3) percentiles; whiskers extend to the most extreme values within 1.5×IQR of the quartiles) **c.** Sequence length distributions initially shown in a, now grouped by the public database the respective sequence originated from. **d.** Shannon entropy distributions initially shown in b now grouped by the public database the respective sequence originated from.

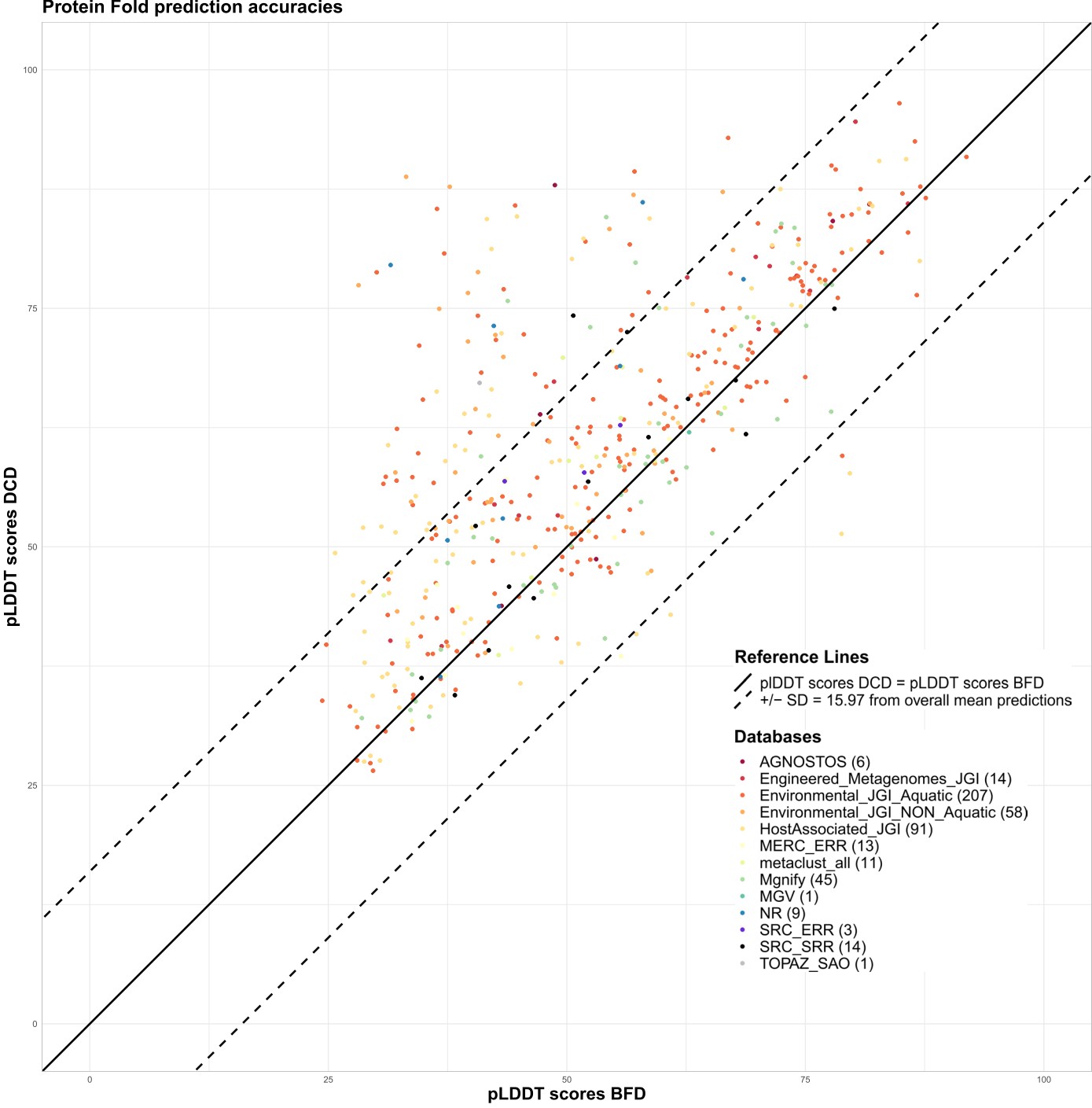

**Extended Data Fig. 4 | AlphaFold2 pLDDT scores of predicted structures using the Big Fantastic Database versus the DIAMOND DeepClust database.** For a sample of 473 cluster representative sequences, we show the pLDDT scores of AlphaFold2 predictions using AlphaFold2's built-in databases BFD and MGnify (x-axis) versus using the DIAMOND DeepClust experimental study database (y-axis). Dashed lines indicate the standard deviation.

# Reporting Summary

## Statistics

For all statistical analyses, confirm that the following items are present in the figure legend, table legend, main text, or Methods section.

| n/a | Confirmed | |
|---|---|---|
| ☐ | ☒ | The exact sample size (*n*) for each experimental group/condition, given as a discrete number and unit of measurement |
| ☒ | ☐ | A statement on whether measurements were taken from distinct samples or whether the same sample was measured repeatedly |
| ☒ | ☐ | The statistical test(s) used AND whether they are one- or two-sided<br>*Only common tests should be described solely by name; describe more complex techniques in the Methods section.* |
| ☒ | ☐ | A description of all covariates tested |
| ☒ | ☐ | A description of any assumptions or corrections, such as tests of normality and adjustment for multiple comparisons |
| ☐ | ☒ | A full description of the statistical parameters including central tendency (e.g. means) or other basic estimates (e.g. regression coefficient) AND variation (e.g. standard deviation) or associated estimates of uncertainty (e.g. confidence intervals) |
| ☒ | ☐ | For null hypothesis testing, the test statistic (e.g. *F*, *t*, *r*) with confidence intervals, effect sizes, degrees of freedom and *P* value noted<br>*Give P values as exact values whenever suitable.* |
| ☒ | ☐ | For Bayesian analysis, information on the choice of priors and Markov chain Monte Carlo settings |
| ☒ | ☐ | For hierarchical and complex designs, identification of the appropriate level for tests and full reporting of outcomes |
| ☒ | ☐ | Estimates of effect sizes (e.g. Cohen's *d*, Pearson's *r*), indicating how they were calculated |

*Our web collection on statistics for biologists contains articles on many of the points above.*

## Software and code

Policy information about availability of computer code

| Data collection | Standard command line tools (wget) |
|---|---|
| Data analysis | ggplot2 3.5.2, custom code available at https://github.com/bbuchfink/deepclust-data |

For manuscripts utilizing custom algorithms or software that are central to the research but not yet described in published literature, software must be made available to editors and reviewers. We strongly encourage code deposition in a community repository (e.g. GitHub). See the Nature Portfolio guidelines for submitting code & software for further information.

## Data

Policy information about availability of data

All manuscripts must include a data availability statement. This statement should provide the following information, where applicable:
- Accession codes, unique identifiers, or web links for publicly available datasets
- A description of any restrictions on data availability
- For clinical datasets or third party data, please ensure that the statement adheres to our policy

The results of clustering 19 billion protein sequences as part of our experimental study can be found at http://www.diamondsearch.org, including FASTA file of representatives, TSV file mapping cluster members to representatives, Parquet files containing all cluster member sequences, index files for fast retrieval and ColabFold database.

April 2023

# Research involving human participants, their data, or biological material

Policy information about studies with human participants or human data. See also policy information about sex, gender (identity/presentation), and sexual orientation and race, ethnicity and racism.

| | |
|---|---|
| Reporting on sex and gender | n/a |
| Reporting on race, ethnicity, or other socially relevant groupings | n/a |
| Population characteristics | n/a |
| Recruitment | n/a |
| Ethics oversight | n/a |

Note that full information on the approval of the study protocol must also be provided in the manuscript.

# Field-specific reporting

Please select the one below that is the best fit for your research. If you are not sure, read the appropriate sections before making your selection.

☒ Life sciences   ☐ Behavioural & social sciences   ☐ Ecological, evolutionary & environmental sciences

For a reference copy of the document with all sections, see nature.com/documents/nr-reporting-summary-flat.pdf

# Life sciences study design

All studies must disclose on these points even when the disclosure is negative.

| | |
|---|---|
| Sample size | For analysis based on subsamples, sample sizes were chosen based on what could be computationally processed in reasonable time. The main benchmark was done using all sequences available in the NCBI nr database at the time and does not include any downsampling. |
| Data exclusions | No data were excluded. |
| Replication | We assume the computations we ran to be deterministic and reproducible and have no reason to believe otherwise. |
| Randomization | n/a |
| Blinding | n/a |

# Reporting for specific materials, systems and methods

We require information from authors about some types of materials, experimental systems and methods used in many studies. Here, indicate whether each material, system or method listed is relevant to your study. If you are not sure if a list item applies to your research, read the appropriate section before selecting a response.

## Materials & experimental systems

| n/a | Involved in the study |
|---|---|
| ☒ ☐ | Antibodies |
| ☒ ☐ | Eukaryotic cell lines |
| ☒ ☐ | Palaeontology and archaeology |
| ☒ ☐ | Animals and other organisms |
| ☒ ☐ | Clinical data |
| ☒ ☐ | Dual use research of concern |
| ☒ ☐ | Plants |

## Methods

| n/a | Involved in the study |
|---|---|
| ☒ ☐ | ChIP-seq |
| ☒ ☐ | Flow cytometry |
| ☒ ☐ | MRI-based neuroimaging |

## Plants

Seed stocks

n/a

Novel plant genotypes

n/a

Authentication

n/a

