## [Peer Review File · Nature Methods]

Clustering the protein universe of life using DIAMOND DeepClust

Corresponding Author: Dr Hajk-Georg Drost

A version of this paper was originally rejected for publication by Nature Methods, however that decision was reconsidered after appeal by the authors.

Version 0:

Decision Letter:

5th May 2023

Dear Hajk,

Your Brief Communication entitled "Sensitive clustering of protein sequences at tree-of-life scale using DIAMOND DeepClust" has now been seen by 3 reviewers, whose comments are attached. While they find your work of potential interest, they have raised serious concerns which in our view are sufficiently important that they preclude publication of the work in Nature Methods, at least in its present form.

As you will see, the reviewers raise concerns about the conceptual and performance advance over existing methods and practical utility of the work.

Should further experimental data allow you to fully address these criticisms we would be willing to look at a revised manuscript (unless, of course, something similar has by then been accepted at Nature Methods or appeared elsewhere). This includes submission or publication of a portion of this work somewhere else. We hope you understand that until we have read the revised paper in its entirety we cannot promise that it will be sent back for peer-review.

If you are interested in revising this manuscript for submission to Nature Methods in the future, please contact me to discuss your appeal before making any revisions. Otherwise, we hope that you find the reviewers' comments helpful when preparing your paper for submission elsewhere.

Sincerely,
Arunima

Arunima Singh, Ph.D.
Senior Editor
Nature Methods

Reviewers' Comments:

Reviewer #1:

Remarks to the Author:

Buchfink et al. present DeepClust, a clustering method accompanied by a large clustered protein sequence database. The authors aim to offer a quick solution for clustering sequences at low sequence identities, applying it to a dataset of 19 billion proteins collected from large databases that were ultimately clustered into 1.7 billion clusters. While fast and scalable clustering algorithms are essential, a more comprehensive evaluation of the method beyond its speed would be valuable.

The clustering method builds upon the CD-HIT or UCLUST clustering approaches, leveraging the speed of DIAMOND v2, published in Nature Methods 2021. DeepClust utilizes these fast search capabilities of DIAMOND v2 for clustering purposes, it

is not entirely clear how this work extends existing cluster methodologies. Dr. Reiter raised a related concern in a community review (<https://www.biorxiv.org/content/10.1101/2023.01.24.525373v1.full>), prompting further clarification on the methodological advancements.

In summary, DeepClust shows promise in terms of its high speed and usability, but a clearer explanation of its methodological novelty and a comparison with previously published databases would better demonstrate the practical advantages of the proposed method and database. Providing more information on the method's unique contributions will help to establish its significance in the field.

Major:

1. I have noticed that Dr. Reiter has provided valuable comments in the bioRxiv community review/Arcadia Science. I would like to ask the author to address these as part of this response or directly within the community review.

2. The abstract's claim that "544 million clusters represent 94% of all known proteins" likely overstates the coverage. With currently 20 petabytes of publically available sequencing data in the SRA, it is likely that still billions of proteins are not covered by the database. For instance, the IMG/M website claims to have over 71 billion genes, nearly four times as much as those clustered in this project (see <https://img.jgi.doe.gov>).

3. The statement "We will make the Experimental Study dataset freely available upon journal publication" is unacceptable. Reviewers must have access to the database during the review process to assess data quality.

4. To better understand the practical utility of the proposed database, it would be useful to compare the performance of the database against the BFD using HHblits in a practical application, such as structure prediction, as mentioned in the manuscript (line 98).

5. The rationale behind performing a unidirectional coverage clustering at 20% sequence identity in the benchmark is unclear. A GO and EC purity-based benchmark might offer better insights into the cluster quality of DeepClust and help determine if fragment clustering at low identities is the appropriate strategy for handling the 19 billion genes. Having a proper quality benchmark would also help to improve Figure 1, which is hard to read and interpret in its current state.

6. To continue the previous point, my intuition tells me that dealing with fragments in the earliest steps followed by a clustering with bi-directional coverage would result in better final clusters. Because each cluster would keep the multi-domain structures and cover a single functional unit rather than a mix of potentially very different proteins.

7. It is commonly accepted ("twilight zone of sequence identity", Rost, <https://academic.oup.com/peds/article/12/2/85/1550637>) that sequence comparison at 20% sequence identity cannot be aligned reliably anymore. Ideally, to establish a clustering with such deep homologies, a benchmark with HMMer3, HHblits or MMseqs2 based profiles should be done. As MMseqs2 is likely the only sufficiently scaling method, it could be used for the clustering benchmark at 20% sequence identity (see <https://github.com/soedinglab/MMseqs2/wiki#how-to-cluster-using-profiles>).

8. A clear explanation of how the 5.5-fold increase in sequence diversity compared to BFD was measured would be helpful. If this is based on a higher number of cluster members, then this would not imply more diversity. In order to see how much of the DeepClust DB is already covered by BFD both databases should be aligned with each other. I would expect that most non-singleton DeepClust clusters will find highly homologous hits in the BFD.

Figure 2 hints to the direction that BFD already covers all the sequence of the DeepClust DB. Also please adjust the colors for "None" and "BFD" to make them more visually distinct and thus enhancing readability. Currently the color scheme makes it nearly impossible to distinguish them.

9. When inferring unknown sequences in UniProt, it is more accurate to use profiles for mapping the cluster back to UniProt or at least all sequences within the cluster, considering the low identity clustering. A relevant work here is <https://elifesciences.org/articles/67667>, which shows that most metagenome sequences can be mapped back using sensitive protein alignment methods like HHblits.

10. Currently, the algorithmic details of DeepClust are not clear when reading in the main manuscript and methods section. Please consider making a main figure explaining it. Especially information on how the cluster time scales with input size and the expected runtime complexity would be valuable, as state-of-the-art methods are approximately linear with input set size.

11. To facilitate reproducibility, please provide open-source scripts and data, such as the materials needed to reproduce Figure 2.

12. The authors justified turning off masking and composition bias in DIAMOND and MMseqs2 to make both methods comparable. Comparing clusterings just by the number of clusters does not make sense if the clustering is meaningless (due to a large number of false positives) due to parameter choices. At 20% sequence identity quality of a clustering is more important than strictly fulfilling the threshold since most residues in the alignment are similar instead of identical, and sequence identity loses significance. An independent measurement would be needed to show quality, e.g., as mentioned above, based on a GO or EC level comparison. Wouldn't this cause a lot of high-scoring false positives?

13. The authors changed the code of the competing method MMseqs2. While the changes might be valuable contributions on their own, this makes the data presented unrealistic from a normal user's point of view, who would never adjust the method in

this way.

Minor:

1. The authors used a ProtT5 language model to analyze the clusters. To do this analysis they must have generated embeddings for each cluster. I would be very interested in how clustering those embeddings would look like.

2. "This striking result shows that while 68% of clusters contain unique sequences, these ~1.16 billion singletons ≥ 5 members Singleton represent only 6% of the 19 billion sequences defining our current protein universe which begs the question whether these distinct proteins are derived from novel orphan genes or whether they represent assembly and annotation artifacts." Other clusterings have previously shown this. The oldest number I can find is from the UniRef 2007 (Suzek et al. 2007) release, where 64% of ~1 million clusters were singletons, while the UniParc contained ~8 million sequences.

3. Discussing the method's limitations and potential future improvements could provide valuable context for readers.

Reviewer #2:

Remarks to the Author:

The authors presented a new version of DIAMOND (DeepClust) for clustering large collections of protein sequences. Benchmarking using NR collection showed that DeepClust gave much faster clustering than the methods it compared to, including MMSeq2 and CD-hit. The authors then applied DeepClust to cluster proteins from the Earth BioGenome project and showed that the 19 billion sequences can be clustered into 1.7 billion clusters of which 32% hold more than one sequence. DeepClust achieved impressive speedup, and it seems to be a useful and needed tool for clustering large collections of sequences.

I only have some small comments.

Using NR database for benchmarking was done using different thresholds 20%, 50% and 90% to show the runtime and number of clusters. I would suggest that the authors add the results for 30% identify, since this is the parameter that was used to cluster the 19 billion sequences from the Earth BioGenome Project.

Need to add a note for Distance EC in Fig. 1 caption. Also need to provide a bit of technical details about the different modes: sensitive, very-sensitive, and ultra-sensitive.

For Fig. 2, the colors for None and BFD are too similar making it hard to distinguish these two. The authors used Fig. 2 to show that "the protein sequence space is dominated by unexplored protein sequences not sufficiently characterized by standard databases". I think this is a bit subjective (also hard to see clearly from the plot). It will be good to see some numbers, for example, how many representatives are not similar to proteins in any of the mentioned collections.

In "In the most sensitive run MMseqs2 still did not discover clusterable homologs for 9.8% of the representatives compared to 2.5% for DIAMOND ultra-sensitive", what are clusterable homologs?

Reviewer #3:

Remarks to the Author:

The authors tackle the problem of clustering very large amounts of protein sequences, a classical problem which remains of high importance given the continued rapid growth in available sequences, and the use of sequence cluster for many high-profile applications, and notably protein structure prediction using deep learning techniques.

The manuscript makes two main contributions. First, it introduces a clustering method called "Diamond DeepClust". Second, it provides a clustering of 19 billion sequences compiled from 15 public sources, mostly from metagenomics.

As I elaborate below in detail, the novelty of the method is not clearly conveyed in the current version of the manuscript. Conceptually, the tool combines the well established Diamond aligner (Buchfink et al Nat Methods 2021) with a clustering approach ("DeepClust") that appears to be conceptually quite similar to Linclust (Steinegger & Söding, Nat Commun 2018). For instance, the idea of cascade clustering and greedy vertex cover algorithm was already introduced by Linclust. Furthermore, the benchmark provided in the paper is very confusing and incomplete.

The 19B-sequence clustering database appears to be the largest such effort to date, and thus constitutes a milestone and a useful resource. However, I am not certain that this, in and of itself, is sufficient to warrant publication in Nature Methods.

Lack of clarity in the presentation of the benchmarks

=====

* The reporting of the benchmarks is confusing. Figure 1 (the main figure on the benchmark) reports time and number of clusters, which says nothing about accuracy. Indeed, the fastest combination of methods reported—Diamond+Linclust—is *not* the one used for the clustering of the 19B sequences. The low informativeness of Figure 1 is evidenced by the fact that the main text repeatedly refers to Supp. Figure 2-7 to justify its claims. At the same time, it is difficult to glean the relevant

information from Supp. Figure 2-7, as each figure measure a partial aspect of accuracy, with the different method combinations appearing in inconsistent order. Instead, I think Figure 1 should summarise the key information, which is not only the speed and number of clusters, but also the accuracy of each variant (e.g. through precision/recall values, F1 measure, AUPRC, or something like that).

* Likewise, the textual description of the benchmark results in the main text is imprecise. Consider for instance the following sentence: "In addition, DIAMOND DeepClust exhibited higher clustering quality as measured by the sensitivity of clusterable homologs found, the completeness of representation by the representative sequences, and the optimality of cluster assignment (Supplementary fig. 2-7)". Which variant of DIAMOND DeepClust does this sentence refer to? (there are multiple combinations of sensitivity levels and target identity levels). "Higher clustering quality" than what method(s) exactly?

* Throughout all benchmark figures, the labels lack clarity. The term "DeepClust" never appears on the figure, and appears to be implicit whenever DIAMOND has no explicit clustering method mentioned. Likewise, whenever MMseqs2 is mentioned without Linclust, it appears that MMseqs's standard clustering approach is used, right? Some terms are not defined in the legend (e.g. "EC" stands for Error Correction). The colour scheme mixes alignment and clustering method. The x-axis (or caption) should specify that time is in wall clock on a 64-core server.

* The confusion around naming extends to the Github code repository, where there is also reference to "diamond cluster" (<https://github.com/bbuchfink/diamond/wiki/Clustering>). How does this differ from DeepClust, if at all?

* Method section, Design subsection: "The benchmarks for clustering at 90% identity were run based on an older version of the NR database downloaded in September 2021 containing 425,032,034 protein sequences and 155,806,124,097 total residues. The database was not hardmasked and no sequences were excluded.". Does this imply that Figure 1 and Supp. Figure 1-7 report results that were performed on inconsistent datasets depending on the method combination? This would add even more confusion to the plots....

* It would be helpful to compare the memory usage for the different combination of tools.

* The "30% sequence identity and 90% coverage threshold" Diamond+DeepClust variant used in the main application does not seem to be included in the benchmarks.

No evidence that DeepClust constitutes an advance over the state-of-the-art

=====
* It appears that Diamond+DeepClust %id=90 is slower than Diamond+Linclust and MMseqs2+Linclust, with negligible improvement in apparent clustering quality (Supplementary figures 1-7). If the improvement is not negligible, the author who should provide clear and compelling evidence of this.

* p2 "...Linclust [is] limited when aiming to cluster billions of proteins with such broad sequence diversity in reasonable time and with sufficient clustering sensitivity at lower identity-boundaries". These claims need to be supported by evidence.

* p5, line 112, "MMseqs2/Linclust ... still suffers from comparatively low performance when clustering at high alignment sensitivity" This is not what I think I see in the benchmarks. Please support this assertion more precisely by referencing the relevant benchmark results.

* Results in the supplementary figures were obtained from a single random selection of 3000 sequences. How robust are these results? Consider repeating this analysis and reporting error bars.

* Page 3: "We further optimized our clustering procedure to ... scale linearly". This claim should be backed by evidence, ideally an empirical figure of CPU time vs number of sequences.

* Besides Linclust, other scalable clustering tools have been introduced in the past few years, such as Rabbitclust and MeShClust3. How do these compare with DeepClust?

* The benchmark exclusively focuses on comparing sequences with cluster representative sequences. How is the relative performance of the various method combinations with respect to non-representative sequences?

Reproducibility

=====

* The command lines for running each tools are provided. Could you please also provide the code to assess the performance and generate the results of Supp. Figure 2-7?

* In the Supplementary Dataset, some values are not known (#NUM!) and there are a few sheets without any description.

Minor points

=====

- * The choice of the name "DeepClust" may be perceived as misleading, since the term "deep" is often associated with "deep learning" these days.
- * In data availability, it is mentioned that this will happen after publication. It would be helpful to specify how these data will be provided, how users will be able to access them, and in which format.
- * In supplementary figure 4, there is no value for cd-hit.
- * Please include a precise definition for sequence coverage.

** For Nature Portfolio general information and news for authors, see <http://npg.nature.com/authors>.

Version 1:

Decision Letter:

16th Jul 2025

Dear Dr. Drost,

Your Brief Communication, "Sensitive clustering of protein sequences at tree-of-life scale using DIAMOND DeepClust", has now been seen by 3 reviewers. As you will see from their comments below, although the reviewers find your work of considerable potential interest, they have raised a few remaining concerns. We are interested in the possibility of publishing your paper in Nature Methods, but would like to consider your response to these concerns before we reach a final decision on publication.

We therefore invite you to revise your manuscript to address these concerns.

Link Redacted

We hope to receive your revised paper within 4 weeks. If you cannot send it within this time, please let us know. In this event, we will still be happy to reconsider your paper at a later date so long as nothing similar has been accepted for publication at Nature Methods or published elsewhere.

OPEN SCIENCE REQUIREMENTS

REPORTING SUMMARY

When revising your manuscript, please update your reporting summary.

For any revision that includes light microscopy data, we ask our authors to please include a completed light microscopy reporting table [https://www.nature.com/documents/Light_microscopy_reporting_table.xlsx] to ensure the methods are described thoroughly. The table will be available to reviewers and ultimately published should the manuscript be accepted at the journal.

IMAGE INTEGRITY

EXTENDED DATA FIGURES

DATA AVAILABILITY

All novel DNA and RNA sequencing data, protein sequences, genetic polymorphisms, linked genotype and phenotype data, gene expression data, macromolecular structures, and proteomics data must be deposited in a publicly accessible database, and accession codes and associated hyperlinks must be provided in the "Data Availability" section.

For papers containing bioimaging data, we strongly recommend depositing the data to Bioimage Archive (<https://www.ebi.ac.uk/bioimage-archive/>). Associated accession codes and hyperlinks should be provided in the "Data Availability" section.

To further increase transparency, we encourage you to provide, in tabular form, the data underlying the graphical representations used in your figures. This is in addition to our data-deposition policy for specific types of experiments and large datasets. For readers, the source data will be made accessible directly from the figure legend. Spreadsheets can be submitted in .xls, .xlsx or .csv formats. Only one (1) file per figure is permitted: thus if there is a multi-paneled figure the source data for each panel should be clearly labeled in the csv/Excel file; alternately the data for a figure can be included in multiple, clearly labeled sheets in an Excel file. File sizes of up to 30 MB are permitted. When submitting source data files with your manuscript

please select the Source Data file type and use the Title field in the File Description tab to indicate which figure the source data pertains to.

CODE AVAILABILITY

Please include a "Code Availability" subsection in the Online Methods which details how your custom code is made available. Only in rare cases (where code is not central to the main conclusions of the paper) is the statement "available upon request" allowed (and reasons should be specified).

MATERIALS AVAILABILITY

SUPPLEMENTARY PROTOCOL

To help facilitate reproducibility and uptake of your method, we ask you to prepare a step-by-step Supplementary Protocol for the method described in this paper. We [encourage authors to share their step-by-step experimental protocols](https://www.nature.com/nature-research/editorial-policies/reporting-standards#protocols) on a protocol sharing platform of their choice and report the protocol DOI in the reference list. Nature Portfolio's protocols.io is a free-to-use and open resource for protocols; protocols deposited onto protocols.io are citable and can be linked from the published article. More details can found at [protocols.io](https://www.protocols.io/help/publish-articles).

ORCID

Sincerely,
Arunima

Arunima Singh, Ph.D.
Senior Editor
Nature Methods

Reviewers' Comments:

Reviewer #1 (Remarks to the Author):

The revised manuscript represents a clear improvement over the previous version. The authors benchmark the most relevant tools including FLSHclust, a recently published method in Science in 2023. The method is technically sound and presents a major improvement in making clustering of large protein sets much more accessible, which are becoming increasingly common (SPIRE, IMG/M, ...). DeepCLUST's scalability across multiple-nodes will allow large scale clustering.

However, the benchmarking and evaluation sections require clarification and revision.

Major points

- The benchmark has improved substantially. In this version, the authors annotated protein sequences with InterPro and generated PFAM architecture strings and use this as ground truth. This is a good benchmark, however, there are still methodological inconsistencies that need to be addressed.

Precision: In the current form it is not clear how the precision is computed, currently it reads "We define the sequence-level precision as the number of sequences in the same cluster of the same domain architecture, divided by the size of the cluster.". How was this counted, was it checked if the representative is consistent with the members or was a majority architecture selected to compute the precision? Please clarify this benchmark.

Sensitivity:

Inconsistent Pfam clan treatment: Precision correctly treats different Pfam families within the same clan as equivalent; however, for sensitivity this is not the case. Please compute this consistently between the two metrics.

Cluster count as recall: An additional recall (sensitivity) independent for the PFAM annotation would be the total number of clusters. The best clustering method should maintain high precision while minimizing the number of clusters. Please include total cluster counts in Figure 1.

Sensitivity ceiling: The current benchmark makes achieving 100% sensitivity impossible due to the fixed sequence-identity threshold. Sequences with identical domain architectures but below this identity cutoff of 30% will be cluster separately. This could be discussed or the definition could be adjusted.

- Figure 2 and "dark" proteins: The current "dark" annotation applies a global 90% query-coverage filter, but most functional annotations result from local alignments that do not meet this threshold. Removing the coverage filter would better reveal the truly dark protein fraction. Additionally, relying solely on UMAP visualization can be misleading, as UMAP often distorts global structure and can produce spurious clusters (e.g., 10.1038/s42003-022-03628-x). To increase interpretability, please consider moving the UMAP to the supplement, or fully replacing it with explicit numerical annotations (counts or percentages) for each class.

- Discussion regarding Greedy vertex cover. While greedy vertex-based clustering can be useful to generate remote clusterings it comes with the risk of being overly sensitive due to wrong edges in the graph. Additionally, it can easily combine groups that should not be merged. It would be great to have this as a strength and limitation discussed.

- Parameter optimization, DeepClust introduces many parameter choices but does not provide sufficient justification for the decisions. E.g. "default uses a search depth of one for the first two rounds of cascaded clustering, and a depth of three for the remaining rounds.", "In addition, we found it advantageous to use a more stringent clustering criterion in earlier rounds of cascaded clustering and relax it gradually in later rounds.". Please explain the rationale behind these settings—were they empirically determined, based on theoretical considerations, or borrowed from previous work?

Minor points

- The manuscript would further improve by including a schematic representation or algorithm that clearly highlights the novelties. While the revisions improved the understandability of the manuscript, some details are still not entirely clear. The manuscript in its current form has enough space to accommodate this addition, which would greatly benefit the reader. I also think that Figures 2 could easily be merged into a single Figure 1, which currently has sufficient available space. Please consider adding this; however, I would leave it to the authors to decide whether or not to implement this suggestion.

Reviewer #1 (Remarks on code availability):

Code is open source and the method is well maintained.

Reviewer #2 (Remarks to the Author):

The authors have spent tremendous efforts revising the manuscript, with more convincing results added to the revision. They have addressed all my comments.

Reviewer #2 (Remarks on code availability):

I did not try to reproduce the results of the paper, as they are based on very large collection of proteins so it would be difficult

even trying to reproduce.

For the github repo, I think the documentation pages can be revised to more clearly show when to use deepclust and how it is different from the other clustering methods ('clust' and 'linclust'). On the wiki page on clustering (<https://github.com/bbuchfink/diamond/wiki/Clustering>), 'deepclust' is hidden deep, and 'diamond deepclust' is not even listed along with 'diamond clust' and 'diamond linclust' among the basic commands.

Reviewer #3 (Remarks to the Author):

The manuscript by Buchfink et al. introduces DIAMOND DeepClust, a method for clustering protein sequences at tree-of-life scale with both high sensitivity and computational efficiency. The core contribution is twofold: (1) a cascaded clustering algorithm that leverages the speed of DIAMOND v2 while preserving sensitivity at low sequence identity thresholds, and (2) the application of this method to generate what is, to my knowledge, the largest and most diverse clustered protein dataset to date, comprising 19 billion sequences drawn from 15 public sources.

When I reviewed the earlier version of this manuscript two years ago, I expressed two major reservations: first, the conceptual novelty of the method over existing tools such as Linclust was not clearly articulated, and second, that the benchmarking was insufficiently rigorous and poorly presented, making it difficult to assess whether DeepClust offered a substantive advance. Nevertheless, I noted that the tool showed potential given its promising performance in our own independent testing.

I am pleased to report that the current revision fully addresses the points raised in my previous report. In particular, the benchmarking has significantly improved: it is now clearer, broader in scope, and better controlled. It includes systematic comparisons across sequence identity regimes and against the most relevant alternatives—MMseqs2-Linclust, MMseqs2-Cluster, and FLSHclust. Most notably, the authors now demonstrate that DeepClust outperforms even MMseqs2-Cluster, which is widely considered a state-of-the-art method, particularly in the low-identity regime that is most relevant for comparative genomics and functional inference across large evolutionary distances.

With the methodological presentation and benchmarking now robust and compelling, I believe this revised manuscript makes a strong case for publication. I just have a couple of minor/discretionary comments below that should be easy to address.

Minor comments

=====

* The authors have removed the results for 90% identity, where DIAMOND DeepClust performs worse in terms of sensitivity. These results could still be reported (e.g. in supplementary) along with a discussion in the main text. The reasoning was that high identity is "not the focus of this work", but a typical user should be aware of this limitation and not assume the package performs best in all scenarios.

* One discretionary suggestion would be to combine Figures 1a and 1b and use a logarithmic scale for CPU hours, although the current format is OK.

* Another discretionary suggestion would be to include a figure that describes the method, as previously suggested by Reviewer 1.

Reviewer #3 (Remarks on code availability):

Code is well documented and could be run without any issue.

Version 2:

Decision Letter:

Our ref: NMETH-BC51641B

7th Oct 2025

Dear Dr. Drost,

Thank you for submitting your revised manuscript "Sensitive clustering of protein sequences at tree-of-life scale using DIAMOND DeepClust" (NMETH-BC51641B). It has now been seen by the original referees and their comments are below. The reviewers find that the paper has improved in revision, and therefore we'll be happy in principle to publish it in Nature Methods, pending minor revisions to satisfy the referees' final requests and to comply with our editorial and formatting guidelines.

TRANSPARENT PEER REVIEW

ORCID

Author names using non-Roman characters

Nature Portfolio journals can support presentation of author names using non-Roman characters in the HTML version of the article. If you wish to, please include author names in parentheses after the Roman-character spelling; [see example online here](https://www.nature.com/articles/s44222-024-00258-2). Currently supported scripts are: Arabic, Chinese, Cyrillic, Devanagari, Greek, Hebrew, Hangeul, Japanese and Persian. You will be asked to verify the rendering is correct at proof stage.

Sincerely,
Arunima

Arunima Singh, Ph.D.
Senior Editor
Nature Methods

Reviewer #1 (Remarks to the Author):

The authors have addressed all my comments.

Version 3:

Decision Letter:

12th Feb 2026

Dear Hajk,

I am pleased to inform you that your Brief Communication, "Clustering the protein universe of life using DIAMOND DeepClust", has now been accepted for publication in Nature Methods. The received and accepted dates will be February 1, 2023 and February 12, 2026. This note is intended to let you know what to expect from us over the next month or so, and to let you know where to address any further questions.

Over the next few weeks, your paper will be copyedited to ensure that it conforms to Nature Methods style. Once your paper is typeset, you will receive an email with a link to choose the appropriate publishing options for your paper and our Author Services team will be in touch regarding any additional information that may be required.

Once proofs are generated, they will be sent to you electronically and you will be asked to send a corrected version within 48 hours. It is extremely important that you let us know now whether you will be difficult to contact over the next month. If this is the case, we ask that you send us the contact information (email, phone and fax) of someone who will be able to check the proofs

and deal with any last-minute problems.

If, when you receive your proof, you cannot meet the deadline, please inform us at rjsproduction@springernature.com immediately.

If you are active on X or Bluesky, please e-mail me your and your coauthors' handles so that we may tag you when the paper is published.

To assist our authors in disseminating their research to the broader community, our SharedIt initiative provides you with a unique shareable link that will allow anyone (with or without a subscription) to read the published article. Recipients of the link with a subscription will also be able to download and print the PDF. As soon as your article is published, you will receive an automated email with your shareable link.

Please note that you and your coauthors may order reprints and single copies of the issue containing your article through Springer Nature Limited's reprint website, which is located at <http://www.nature.com/reprints/author-reprints.html>. If there are any questions about reprints please send an email to author-reprints@nature.com and someone will assist you.

Best regards,
Arunima

Arunima Singh, Ph.D.
Senior Editor
Nature Methods

** Visit the Springer Nature Editorial and Publishing website at http://www.springernature.com/editorial-and-publishing-jobs?utm_source=ejP_NMeth_email&utm_medium=ejP_NMeth_email&utm_campaign=ejp_Nmeth for more information about our career opportunities. If you have any questions please click [here](mailto:editorial.publishing.jobs@springernature.com).**

Dear reviewers,

We would like to express our sincere gratitude for carefully reading our manuscript and for the excellent suggestions that allowed us to significantly improve the communication of our Method to a broad life science audience. We took the extensive feedback as an opportunity to substantially extend our method beyond the original submission.

As suggested by reviewer #1, our revised benchmark now focuses on clustering with bi-directional sequence coverage. We also replaced the alignment-based evaluation by an annotated ground truth using Pfam domains. While this benchmark design also has certain limitations, it should represent a biologically informed evaluation and address the issues raised by the reviewers with respect to our previous benchmark.

As conflating clustering runs at 90%, 50% and 20% sequence identity thresholds into the same benchmark clearly caused some confusion, we now only show results for deep clustering (20% identity). This keeps the benchmark presentation concise and easily interpretable. We consider deep clustering the most challenging and interesting problem to solve and therefore chose it as the focus of this work.

We did not repeat the experimental study clustering run that comprised 19 billion sequences with different parameters. This run was carried out using 30% sequence identity and 90% uni-directional sequence coverage thresholds to match the settings used to create the Big Fantastic Database and provide a comparable result (in terms of cluster count). It would be possible to repeat this computation using bi-directional coverage, a different identity threshold and also ingesting more data, but we would like to receive some guidance from the reviewers with respect to this point.

Sincerely yours,

Hajk-Georg Drost

On behalf of all the authors.

Detailed Response to Reviewers

Reviewer #1:

Buchfink et al. present DeepClust, a clustering method accompanied by a large clustered protein sequence database. The authors aim to offer a quick solution for clustering sequences at low sequence identities, applying it to a dataset of 19 billion proteins collected from large databases that were ultimately clustered into 1.7 billion clusters. While fast and scalable clustering algorithms are essential, a more comprehensive evaluation of the method beyond its speed would be valuable.

The clustering method builds upon the CD-HIT or UCLUST clustering approaches, leveraging the speed of DIAMOND v2, published in Nature Methods 2021. DeepClust utilizes these fast search capabilities of DIAMOND v2 for clustering purposes, it is not entirely clear how this work extends existing cluster methodologies. Dr. Reiter raised a related concern in a community review (<https://www.biorxiv.org/content/10.1101/2023.01.24.525373v1.full>), prompting further clarification on the methodological advancements.

In summary, DeepClust shows promise in terms of its high speed and usability, but a clearer explanation of its methodological novelty and a comparison with previously published databases would better demonstrate the practical advantages of the proposed method and database. Providing more information on the method's unique contributions will help to establish its significance in the field.

Response: We would like to thank this reviewer for taking the time to carefully assess our methodology and for providing such constructive feedback which allowed us to significantly improve our manuscript, especially regarding a clearer explanation of our methodological novelty.

Major:

1. I have noticed that Dr. Reiter has provided valuable comments in the bioRxiv community review/Arcadia Science. I would like to ask the author to address these as part of this response or directly within the community review.

Response: As suggested by this reviewer, we now address the bioRxiv comments in a point-by-point response letter which is attached to this letter as Supplementary Comments.

2. The abstract's claim that "544 million clusters represent 94% of all known proteins" likely overstates the coverage. With currently 20 petabytes of publically available sequencing data in the SRA, it is likely that still billions of proteins are not covered by the database. For instance, the IMG/M website claims to have over 71 billion genes, nearly four times as much as those clustered in this project (see <https://img.jgi.doe.gov>).

Response: We completely agree that this statement is prone to cause a misunderstanding. We meant 94% of all known predicted, assembled, and non-embargoed proteins (about 50% of proteins at the IMG/M website were under embargo and could not be used for our publication, the remainder of what we excluded were fragments of 50aa or less), which amounted to about 15 billion proteins at the IMG/M website at the time we downloaded the data (March 2022). As of May 2025, there are 44.6 billion unrestricted proteins available at IMG/M, or 69.2 billion when also including unassembled proteins (which are mostly fragments of 50aa or less). It would be possible to recompute our clustering to include all of this data, but we would like to receive some guidance from the reviewers with respect to this point.

To avoid confusing readers, we now change this statement to "544 million clusters represent 94% of our 19 billion clustered proteins."

3. The statement "We will make the Experimental Study dataset freely available upon journal publication" is unacceptable. Reviewers must have access to the database during the review process to assess data quality.

Response: We apologize for this shortfall. We provided all computationally reproducible command line calls to reproduce our result dataset within this submission. We now deposited our dataset at <https://objectstore.hpcloud.mpcdf.mpg.de/deepclust/index.html> with a dedicated data retrieval system (https://github.com/drostlab/deepclust_dataretrieval) and added the link to our revised manuscript (Data Availability).

4. To better understand the practical utility of the proposed database, it would be useful to compare the performance of the database against the BFD using HHblits in a practical application, such as structure prediction, as mentioned in the manuscript (line 98).

Response: This is an excellent suggestion. We addressed this point (Methods, section: Structure Prediction) and showed that our clustered database is able to improve AlphaFold2 structure predictions for sequences not sufficiently represented in smaller databases. See also new Suppl. Fig. 5.

This question was also comprehensively investigated by DeepMSA2 (Zheng et al., Nature Methods 2024), showing that searching through databases of up to 40 billion sequences significantly improves structure prediction. Deep clustering contributes to this application by compressing such datasets to a manageable scale and reducing the computational burden of such computations.

5. The rationale behind performing a unidirectional coverage clustering at 20% sequence identity in the benchmark is unclear. A GO and EC purity-based benchmark might offer better insights into the cluster quality of DeepClust and help determine if fragment clustering at low identities is the appropriate strategy for handling the 19 billion genes. Having a proper quality benchmark would also help to improve Figure 1, which is hard to read and interpret in its current state.

Response: Our rationale is to cluster as sensitively as possible on the pairwise local alignment level. Independent verification of the clustering quality by Pfam annotations now shows a precision of >95%, so it would not seem necessary to reduce sensitivity by setting a higher sequence identity threshold. Our revised benchmark now uses bi-directional coverage instead of unidirectional coverage clustering, ensuring high precision while maintaining high sensitivity levels. We opted to use Pfam instead of GO or EC annotations as homology can be measured more precisely and allows us to quantify both precision and sensitivity of the clustering, while GO consistency or EC purity are only measures of precision. It would be possible to provide these as additional benchmarks if requested by the reviewer.

6. To continue the previous point, my intuition tells me that dealing with fragments in the earliest steps followed by a clustering with bi-directional coverage would result in better final clusters. Because each cluster would keep the multi-domain structures and cover a single functional unit rather than a mix of potentially very different proteins.

Response: To directly address and quantify this point our revised benchmark now uses bi-directional coverage. We also show results for uni-directional coverage clustering which indeed confirms the reviewers intuition of bi-directional coverage being superior in this regard.

7. It is commonly accepted (“twilight zone of sequence identity”, Rost, <https://academic.oup.com/peds/article/12/2/85/1550637>) that sequence comparison at 20% sequence identity cannot be aligned reliably anymore. Ideally, to establish a clustering with such deep homologies, a benchmark with HMMer3, HHblits or MMseqs2 based profiles should be done. As MMseqs2 is likely the only sufficiently scaling method,

it could be used for the clustering benchmark at 20% sequence identity (see <https://github.com/soedinglab/MMseqs2/wiki#how-to-cluster-using-profiles>).

Response: To our knowledge, (Rost, 1999) used the distance to the HSP curve as a measure of significance for alignments, without using stringent e-value and sequence coverage thresholds, or repeat masking and compositional score correction. When using such filters, pairwise alignments are quite reliable down to 20% identity. This is confirmed for our revised benchmark by Pfam annotations showing >95% precision of such clustering.

We attempted to run the suggested computation of MMseqs2 for the NR database on a server using 64 AMD EPYC cores. The computation expired due to a 3 week time limit. According to the command line output, it was about 25% complete at this point. We documented this run in the supplementary information.

8. A clear explanation of how the 5.5-fold increase in sequence diversity compared to BFD was measured would be helpful. If this is based on a higher number of cluster members, then this would not imply more diversity. In order to see how much of the DeepClust DB is already covered by BFD both databases should be aligned with each other. I would expect that most non-singleton DeepClust clusters will find highly homologous hits in the BFD. Figure 2 hints to the direction that BFD already covers all the sequence of the DeepClust DB. Also please adjust the colors for "None" and "BFD" to make them more visually distinct and thus enhancing readability. Currently the color scheme makes it nearly impossible to distinguish them.

Response: Diversity can be understood as (A) the amount of variation within a fixed set of protein families or (B) the total number of distinct protein families. Diversity in the sense of (A) is certainly not biologically irrelevant. For example, species-level diversity is usually measured by clustering at 90% identity. Compared to this, our clustering criterion (30% identity, 90% coverage) results in a much lower estimate of diversity. Further, applications such as structure prediction also rely on the variation observed within protein families. The cluster count measures diversity with respect to the clustering criterion used to create the clustered database. In both cases, for the construction of the Big Fantastic Database (AlphaFold2) and our own database (DeepClust), 30% identity and 90% uni-directional coverage were chosen as clustering criteria.

To measure diversity in the sense of (B), we have now aligned a sample of our non-singleton representative sequences against the Big Fantastic Database using HHblits and present the results as Supplementary Figure 4. When requiring at least 60% sequence

coverage, 72.5% of our non-singleton (cluster size ≥ 3) representatives can be mapped to a BFD sequence. Projected to our whole dataset consisting of 339 million clusters with a size of at least three, this would correspond to 93 million novel sequence families not represented in the BFD.

We have also changed the color scheme of the figure to make it more readable.

9. When inferring unknown sequences in UniProt, it is more accurate to use profiles for mapping the cluster back to UniProt or at least all sequences within the cluster, considering the low identity clustering. A relevant work here is <https://elifesciences.org/articles/67667>, which shows that most metagenome sequences can be mapped back using sensitive protein alignment methods like HHblits.

Response: We have mapped a sample of our representatives against the BFD using HHblits (see last point). Since the BFD includes UniProt (Jumper et al., 2021), this point should be addressed by this analysis.

10. Currently, the algorithmic details of DeepClust are not clear when reading in the main manuscript and methods section. Please consider making a main figure explaining it. Especially information on how the cluster time scales with input size and the expected runtime complexity would be valuable, as state-of-the-art methods are approximately linear with input set size.

Response: This is an excellent comment and we now incorporated the suggested scaling plot as Fig. 1a and 1b. With respect to linear scaling, the reviewer refers to MMseqs2/LinClust (Steinegger et al. 2018). We note that the LinClust algorithm is only suitable for clustering similar sequences at high sequence identities. The LinClust paper shows a benchmark for clustering at 90%, 70%, and 50% sequence identity, while the latter two already have substantially reduced sensitivity (see Fig. 3 of the LinClust paper). LinClust is not sufficiently sensitive to cluster at low sequence identity levels (which is also shown by our revised benchmark), and the regular MMseqs2 clustering workflow does not have linear scaling, as shown by our Fig. 1a. For example, while 55 million sequences can be clustered in 18h, 546 million sequences (NR database) require about 4 weeks.

The cascaded clustering algorithm as implemented by DIAMOND DeepClust and MMseqs2 can be expected to have roughly quadratic run time in the number of clusters (not the number of input sequences), as the run time should be dominated by all-vs-all alignment in the most sensitive mode.

We have restructured the methods section and made it more detailed for more clarity with respect to the algorithm.

11. To facilitate reproducibility, please provide open-source scripts and data, such as the materials needed to reproduce Figure 2.

Response: We now include all scripts in the revised manuscript.

12. The authors justified turning off masking and composition bias in DIAMOND and MMseqs2 to make both methods comparable. Comparing clusterings just by the number of clusters does not make sense if the clustering is meaningless (due to a large number of false positives) due to parameter choices. At 20% sequence identity quality of a clustering is more important than strictly fulfilling the threshold since most residues in the alignment are similar instead of identical, and sequence identity loses significance. An independent measurement would be needed to show quality, e.g., as mentioned above, based on a GO or EC level comparison. Wouldn't this cause a lot of high-scoring false positives?

Response: For our revised benchmark, high precision of the clustering (>95%) is now confirmed by independent verification using Pfam annotations (see Fig. 1c). The tools were run with masking and compositional bias correction enabled in this revised benchmark.

13. The authors changed the code of the competing method MMseqs2. While the changes might be valuable contributions on their own, this makes the data presented unrealistic from a normal user's point of view, who would never adjust the method in this way.

Response: To avoid any misunderstanding, we now use an unmodified version of MMseqs2 for the new benchmarks we present in this revised version of our manuscript.

Minor:

1. The authors used a ProtT5 language model to analyze the clusters. To do this analysis they must have generated embeddings for each cluster. I would be very interested in how clustering those embeddings would look like.

Response: We did not create embeddings for each cluster, but only for a sample of 1 million representative sequences to illustrate further downstream applications of

DeepClust. We assume that a clustering of the embeddings for this sample is only of limited interest in the context of our methods article.

2. “This striking result shows that while 68% of clusters contain unique sequences, these ~1.16 billion singletons ≥ 5 members Singleton represent only 6% of the 19 billion sequences defining our current protein universe which begs the question whether these distinct proteins are derived from novel orphan genes or whether they represent assembly and annotation artifacts.”. Other clusterings have previously shown this. The oldest number I can find is from the UniRef 2007 (Suzek et al. 2007) release, where 64% of ~1 million clusters were singletons, while the UniParc contained ~8 million sequences.

Response: We agree it is an excellent idea to place our findings in agreement with previous observations which showed similar proportions. We now write: “This striking result shows that while 68% of clusters contain unique sequences, these ~1.16 billion singletons represent only 6% of the 19 billion sequences defining our current protein universe which begs the question whether these distinct proteins are derived from novel orphan genes or whether they represent assembly and annotation artifacts. These proportions between singletons and non-singletons are in agreement with previous clustering efforts (Suzek et al. 2007, Jumper et al. 2021), thereby illustrating that the proportion of singletons does not change much when adding more protein sequences from newly sequenced organisms.”.

3. Discussing the method's limitations and potential future improvements could provide valuable context for readers.

Response: This is an excellent suggestion. We now include a new section “Limitations of DIAMOND DeepClust and Future Improvements” in the Methods section.

Reviewer #2:

The authors presented a new version of DIAMOND (DeepClust) for clustering large collections of protein sequences. Benchmarking using NR collection showed that DeepClust gave much faster clustering than the methods it compared to, including MMSeq2 and CD-hit. The authors then applied DeepClust to cluster proteins from the Earth BioGenome project and showed that the 19 billion sequences can be clustered into 1.7 billion clusters of which 32% hold more than one sequence. DeepClust achieved impressive speedup, and it seems to be a useful and needed tool for clustering large collections of sequences.

I only have some small comments.

Response: We would like to thank this reviewer for taking the time to carefully review our manuscript and for recognizing that deep clustering the Tree of Life at sufficient computational speed allows significant compression of the protein sequence space for further molecular research.

Using NR database for benchmarking was done using different thresholds 20%, 50% and 90% to show the runtime and number of clusters. I would suggest that the authors add the results for 30% identify, since this is the parameter that was used to cluster the 19 billion sequences from the Earth BioGenome Project.

Response: To keep the presentation of the main benchmark succinct, our revised benchmark only contains results for clustering at 20% identity. We used the 30% identity threshold for clustering the 19 billion sequences to make the result directly comparable to the Big Fantastic Database which used the same threshold. We carried out a benchmarking run using the 30% identity setting and included the results in the method section (chapter “Benchmark design”).

Need to add a note for Distance EC in Fig. 1 caption. Also need to provide a bit of technical details about the different modes: sensitive, very-sensitive, and ultra-sensitive.

Response: A detailed description and benchmark of the different sensitivity modes is available in the DIAMOND v2 paper (Buchfink et al., 2021). This directly applies to clustering as well. We made this point clear by adding the following sentence to the Methods section of the manuscript: “The different sensitivity modes available for DIAMOND DeepClust runs correspond to the alignment sensitivity modes introduced in DIAMOND v2 (Buchfink et al., 2021).”.

For Fig. 2, the colors for None and BFD are too similar making it hard to distinguish these two. The authors used Fig. 2 to show that “the protein sequence space is dominated by unexplored protein sequences not sufficiently characterized by standard databases”. I think this is a bit subjective (also hard to see clearly from the plot). It will be good to see some numbers, for example, how many representatives are not similar to proteins in any of the mentioned collections.

Response: We apologize for this poor labeling and choice of color codes. We now address these points and present a refined Figure 2 and more detailed quantification in the figure caption.

In “In the most sensitive run MMseqs2 still did not discover clusterable homologs for 9.8% of the representatives compared to 2.5% for DIAMOND ultra-sensitive”, what are clusterable homologs?

Response: By “clusterable homologs” we meant pairs of sequences that satisfy the clustering criterion, 80% coverage of the cluster member sequence and optionally a sequence identity threshold in our case.

Reviewer #3:

The authors tackle the problem of clustering very large amounts of protein sequences, a classical problem which remains of high importance given the continued rapid growth in available sequences, and the use of sequence cluster for many high-profile applications, and notably protein structure prediction using deep learning techniques.

The manuscript makes two main contributions. First, it introduces a clustering method called “Diamond DeepClust”. Second, it provides a clustering of 19 billion sequences compiled from 15 public sources, mostly from metagenomics.

Response: We would like to thank this reviewer for the opportunity to incorporate these constructive suggestions into a revised version of our manuscript. All suggestions were clearly meant to improve our methodology for the high quality standards of *Nature Methods* and we address all points in full to match these expectations.

As I elaborate below in detail, the novelty of the method is not clearly conveyed in the current version of the manuscript. Conceptually, the tool combines the well established Diamond aligner (Buchfink et al Nat Methods 2021) with a clustering approach (“DeepClust”) that appears to be conceptually quite similar to Linclust (Steinegger & Söding, Nat Commun 2018). For instance, the idea of cascade clustering and greedy vertex cover algorithm was already introduced by Linclust. Furthermore, the benchmark provided in the paper is very confusing and incomplete.

Response: While our basic approach is similar to that of MMseqs2/Linclust, we have also refined and significantly extended it in many ways (especially in the context of bi-directional coverage clustering, and making linear-time clustering more sensitive and scalable, see methods section). Our extensive efforts resulted in a 35-fold faster computational speedup. We believe that this achievement was only possible due to new algorithmic and software engineering approaches that we outline in detail in the methods section.

The 19B-sequence clustering database appears to be the largest such effort to date, and thus constitutes a milestone and a useful resource. However, I am not certain that this, in and of itself, is sufficient to warrant publication in Nature Methods.

Response: We very much appreciate this fair discussion and hope that the past two years have demonstrated that the ability of clustering tens of billions of sequences with various clustering modes/sensitivity/criteria can unlock exciting new downstream research as was illustrated by the studies by Zheng et al (2024), Nature Methods (DeepMSA), Barrio-Hernandez et al. (2023), Nature (AFDB structure clustering), Pavlopoulos et al. (2023), Nature (HipMCL), and Altae-Tran et al. (2023), Science (FLSHclust). Our intention with this method paper is to equip any user able to operate a command line tool with a protein sequence clustering approach that is able to scale to the tens of billions of proteins currently available in public databases and the hundreds of billions of protein sequences projected to exist within this decade. We took this comment seriously and now show that DIAMOND DeepClust can outperform all high-profile clustering studies.

Lack of clarity in the presentation of the benchmarks

=====

* The reporting of the benchmarks is confusing. Figure 1 (the main figure on the benchmark) reports time and number of clusters, which says nothing about accuracy. Indeed, the fastest combination of methods reported—Diamond+Linclust—is *not* the one used for the clustering of the 19B sequences. The low informativeness of Figure 1 is evidenced by the fact that the main text repeatedly refers to Supp. Figure 2-7 to justify its claims. At the same time, it is difficult to glean the relevant information from Supp. Figure 2-7, as each figure measure a partial aspect of accuracy, with the different method combinations appearing in inconsistent order. Instead, I think Figure 1 should summarise the key information, which is not only the speed and number of clusters, but also the accuracy of each variant (e.g. through precision/recall values, F1 measure, AUPRC, or something like that).

Response: Figure 1 now contains all relevant information for our revised benchmark (see also response to Reviewer #1), including sensitivity and precision of the clustering, as well as runtimes and scaling.

* Likewise, the textual description of the benchmark results in the main text is imprecise. Consider for instance the following sentence: "In addition, DIAMOND DeepClust exhibited higher clustering quality as measured by the sensitivity of clusterable homologs found,

the completeness of representation by the representative sequences, and the optimality of cluster assignment (Supplementary fig. 2-7)". Which variant of DIAMOND DeepClust does this sentence refer to? (there are multiple combinations of sensitivity levels and target identity levels). "Higher clustering quality" than what method(s) exactly?

Response: This statement referred to the deep clustering runs (20% identity) in sensitive and very-sensitive mode of DIAMOND DeepClust, showing higher clustering quality than the respective runs of MMseqs2. These benchmark metrics are no longer applicable in the context of our revised benchmark.

* Throughout all benchmark figures, the labels lack clarity. The term "DeepClust" never appears on the figure, and appears to be implicit whenever DIAMOND has no explicit clustering method mentioned. Likewise, whenever MMseqs2 is mentioned without Linclust, it appears that MMseq's standard clustering approach is used, right? Some terms are not defined in the legend (e.g. "EC" stands for Error Correction). The colour scheme mixes alignment and clustering method. The x-axis (or caption) should specify that time is in wall clock on a 64-core server.

Response: We have rephrased the text with respect to these points to provide more clarity. We now write "MMseqs2" when we refer to the standard clustering mode of MMseqs2 (corresponding to the command "mmseqs cluster") and MMseqs2/Linclust when referring to the Linclust mode (command "mmseqs linclust").

* The confusion around naming extends to the Github code repository, where there is also reference to "diamond cluster" (<https://github.com/bbuchfink/diamond/wiki/Clustering>). How does this differ from DeepClust, if at all?

Response: The command line "diamond cluster" also refers to the DIAMOND DeepClust software. While DIAMOND DeepClust was designed with the purpose to efficiently cluster at low identity levels, users can also employ DeepClust with higher identity levels (i.e. >20%). This is why we chose to name the command line flag "cluster" but the name of our software DeepClust.

* Method section, Design subsection: "The benchmarks for clustering at 90% identity were run based on an older version of the NR database downloaded in September 2021 containing 425,032,034 protein sequences and 155,806,124,097 total residues. The database was not hardmasked and no sequences were excluded.". Does this imply that Figure 1 and Supp. Figure 1-7 report results that were performed on inconsistent datasets

depending on the method combination? This would add even more confusion to the plots....

Response: This issue is resolved in our revised benchmark.

* It would be helpful to compare the memory usage for the different combination of tools.

Response: We now provide a table of memory usage in the supplementary information (supplementary table 1).

* The "30% sequence identity and 90% coverage threshold" Diamond+DeepClust variant used in the main application does not seem to be included in the benchmarks.

Response: These parameter settings were chosen to make the results directly comparable to the Big Fantastic Database (introduced in the AlphaFold2 paper), not necessarily because we consider them optimal. To keep the presentation of the main benchmark succinct, we did not include a run using these settings into Fig. 1. We conducted a corresponding run and included the results in the methods section (chapter "Benchmark design").

No evidence that DeepClust constitutes an advance over the state-of-the-art

=====

* It appears that Diamond+DeepClust %id=90 is slower than Diamond+Linclust and MMseqs2+Linclust, with negligible improvement in apparent clustering quality (Supplementary figures 1-7). If the improvement is not negligible, the author who should provide clear and compelling evidence of this.

Response: We believe the reviewer may have confused runs with and without the Linclust option of MMseqs2 and DIAMOND enabled, which are not directly comparable due to the reduced clustering sensitivity when using the Linclust option. More importantly, the statement about lack of advance seems to be based on considering clustering runs at 90% sequence identity only. Clustering highly similar sequences at 90% identity is a comparatively easy problem to solve, and not the focus of this work. DIAMOND DeepClust is focused on deep clustering down to e.g. 20% identity, detecting distant protein relatives across large evolutionary distances, which is a much more complex problem to solve. To make this point clear, our revised benchmark is now limited to deep clustering.

* p2 "...Linclust [is] limited when aiming to cluster billions of proteins with such broad sequence diversity in reasonable time and with sufficient clustering sensitivity at lower identity-boundaries". These claims need to be supported by evidence.

Response: The limitation refers to sensitivity with respect to MMseqs2/Linclust, as Linclust is not suitable for deep clustering at low sequence identity levels. With respect to MMseqs2, this refers to a limitation of performance and scalability. MMseqs2 runs for >3 weeks for clustering the NR database (546 million sequences) with a highly non-linear scaling behavior (Fig. 1a), limiting its ability to scale to future dataset sizes of tens or hundreds of billions of sequences.

* p5, line 112, "MMseqs2/Linclust ... still suffers from comparatively low performance when clustering at high alignment sensitivity" This is not what I think I see in the benchmarks. Please support this assertion more precisely by referencing the relevant benchmark results.

Response: This statement referred to DIAMOND DeepClust running 11-fold to 13-fold faster than MMseqs2 for deep clustering in our original benchmark. This speedup has now improved to 35-fold in our revised software and benchmark (Fig. 1a).

* Results in the supplementary figures were obtained from a single random selection of 3000 sequences. How robust are these results? Consider repeating this analysis and reporting error bars.

Response: We did include confidence intervals for these numbers in the source data of these figures confirming that this sample size is sufficient. This is not applicable to the revised Pfam-based benchmark, as the evaluation covers the whole dataset, not just a sample.

* Page 3: "We further optimized our clustering procedure to ... scale linearly". This claim should be backed by evidence, ideally an empirical figure of CPU time vs number of sequences.

Response: This statement refers to linear scaling to multiple compute nodes in an HPC environment, which we have shown in our previous DIAMOND v2 paper (Buchfink et al., 2021). Scaling of runtime in the number of sequences is now empirically shown in Fig. 1a and 1b.

* Besides Linclust, other scalable clustering tools have been introduced in the past few years, such as Rabbitclust and MeShClust3. How do these compare with DeepClust?

Response: We would like to point out that both RabbitTClust and MeShClust v3.0 are tools for clustering DNA sequences only, hence a comparison to DIAMOND DeepClust which is focused on clustering protein sequences is not applicable.

* The benchmark exclusively focuses on comparing sequences with cluster representative sequences. How is the relative performance of the various method combinations with respect to non-representative sequences?

Response: Our revised benchmark evaluates the clustering quality on the level of the input sequences only, most of which are non-representative sequences.

Reproducibility =====

* The command lines for running each tools are provided. Could you please also provide the code to assess the performance and generate the results of Supp. Figure 2-7?

Response: We now provide this code in a GitHub repository which can be publicly accessed at <https://github.com/bbuchfink/deepclust-data>

* In the Supplementary Dataset, some values are not known (#NUM!) and there are a few sheets without any description.

Response: The tables referred here no longer exist in our revision.

Minor points =====

* The choice of the name “DeepClust” may be perceived as misleading, since the term “deep” is often associated with “deep learning” these days.

Response: While we understand the concerns of Reviewer #3, we would like to point out that the purpose of our software is to perform deep sequence clustering at low identity levels. The term “deep” refers to low identity clustering “deep into the tree of life”. While we appreciate that “deep learning” is a prominent term these days, we prefer to keep our name under the assumption that even the “deep learning” hype will be less pronounced at some point.

* In data availability, it is mentioned that this will happen after publication. It would be helpful to specify how these data will be provided, how users will be able to access them, and in which format.

Response: The data has now been made available (see Data Availability).

* In supplementary figure 4, there is no value for cd-hit.

Response: This is no longer applicable.

* Please include a precise definition for sequence coverage.

Response: This has now been included in the methods section (chapter “Representative-based clustering”).

Supplementary Comments (in response to bioRxiv request of Reviewer #1)

>We present DIAMOND DeepClust, an ultra-fast cascaded clustering method optimized to cluster the 19 billion protein sequences currently defining the protein biosphere

Arcadia: *It would be super helpful here to point out where these protein sequences come from -- NCBI nr, mmseqs sets, etc.*

Response: This information is contained in the supplementary information.

>we identified the ability to cluster this vast protein sequence diversity space as a key factor currently limiting the association of sequences across large sets of divergent species

Arcadia: *Can you add a few more details on how you identified this? why are tools in mmseqs2 not sufficient here? what innovation is needed to overcome whatever barriers exist?*

Response: MMseqs2 is comparatively slower than DIAMOND DeepClust, limiting its ability to scale to future dataset sizes of many billions of sequences (see Fig. 1a).

species

Arcadia: *is each of the 1.8 million genomes to come from a separate species, or will some be different strains of the same species?*

Response: The Earth BioGenome Project aims to sequence 1.8 million different species.

Current protein clustering approaches implemented in the standard tools CD-hit, UClust, and Linclust are limited when aiming to cluster billions of proteins with such broad sequence diversity in reasonable time and with sufficient clustering sensitivity at lower identity-boundaries.

Arcadia: *What are the limitations? why won't something like linclust work here?*

Response: Answered above in response to reviewer #3.

> 19 billion sequences

Arcadia: *What are the database sources for these sequences? are they only eukaryotic, or do they include bacteria and archaea too?*

Response: This information is contained in the supplementary information. The majority of the sequences are bacterial.

> 18 days on 27 high

Arcadia: *what was the RAM usage?*

Response: It may have used up to 2TB of RAM.

> Finally, we designed a re-clustering procedure that allows users to add new sequences to a large collection of existing clusters so that the sequencing and assembly community can swiftly add incoming sequences to our biosphere cluster database without the need to re-cluster the entire dataset (Methods)

Arcadia: *How does this impact cluster membership? this feels akin to the 16s debate of OTUs vs. ASVs*

Response: Adding sequences to a clustering this way does not affect the cluster membership of existing sequences.

> 30% sequence identity

Arcadia: *Why was this sequence identity selected?*

Response: The same threshold was used for creating the Big Fantastic Database, we wanted to keep our result directly comparable.

> these ~1.16 billion unique sequences comprise only ~6% of the full set of 19billion sequences

Arcadia: *Do they come from weird taxonomies too? or metagenomes or something?*

Response: We are not sure what “weird” taxonomies are but the lion’s share of these sequences come from metagenomes.

> 18.1 million CPU hours compared to 194 million CPU hours with MMSeqs2 which makes this computation feasible today on existing HPC systems (Methods)

Arcadia: *This doesn't feel like that big of a difference...yes, mmseqs2 would take 10x as long, but still feels like it could be accomplished on current compute infrastructure. If that's not true, I think it would be beneficial to highlight that.*

Response: The cost of 194 million CPU hours would be about 8 million USD on a cloud computing service, which is certainly not negligible. The cost difference would be more pronounced for bi-directional coverage clustering due to the higher speedup achieved by DIAMOND DeepClust.

> *can be compressed into 335 million centroids for downstream analyses (Supplementary fig. 10)*

Arcadia: *It might be good to highlight that this is on the same order as the current size of NCBI nr (or at least I think it is), meaning our algorithms can already handle searches at this scale*

Response: Ok.

> *Experimental Study*

Arcadia: *but this doesn't exist yet right? I think it would be good to clarify that here*

Response: The experimental study has been done and the results exist.

> *Data availability*

Arcadia: *I don't see this section in the preprint, are the databases available now?*

Response: We have made the data available now (see Data Availability).

> *Although MMSeqs2/LinClust15 presented a considerable advancement over Cd-hit and UClust, it still suffers from comparatively low performance when clustering at high alignment sensitivity, thereby introducing an analytics bottleneck when attempting to scale to >27 billion estimated*

Arcadia: *Less but why? what is your methodological advancement that overcomes this?*

Response: The advancement is the higher base performance of the DIAMOND aligner combined with our algorithmic improvements of the clustering procedure (see Methods section).

> *Fig. 1*

Arcadia: *is it possible to increase the font text size in this figure? it is very difficult to read*

Response: The figure has been removed in this revision and the new version has now improved font size.

> clusterable homologs found

Arcadia: *Where are the gold standard set of homologs defined?*

Response: This was defined based on the clustering criterion in our original benchmark. We now use the Pfam domain architecture to annotate the ground truth in our revised benchmark.

> optimality of cluster assignment

Arcadia: *How is this calculated?*

Response: The optimal assignment was given by the representative sequence whose alignment against the member sequence has the lowest e-value while fulfilling the clustering criterion. This metric is no longer applicable for our revised benchmark.

> Fig. 2

Arcadia: *Can you increase the font size for panels C and D and the y axis on panels A and B?*

Response: We will increase the font sizes accordingly.

> In the first round, we subsample the seed space using minimizers with a window size of 12 which we empirically found to provide a good balance between speed and sensitivity, and attempt to achieve linear computational scaling of comparisons by considering only seed hits against the longest sequence for identical seeds rather than trialing all possible combinations

Arcadia: *Did you consider using UKHS or something like it here?
<https://kingsfordlab.cbd.cmu.edu/publication/orenstein-2016-compactkmers/>*

Response: No but it could be an interesting future improvement.

> Supplementary fig. 1

Arcadia: *Can you make the fonts larger in these panels?*

Response: The figure has been removed.

> We hard-masked this database using tantan with default settings and removed all sequences that were masked over >10% of their range, resulting in a reduced database of 445,610,930 sequences.

Arcadia: What types of sequences did this remove? what biological biases are introduced here over using full nr?

Response: That is hard to answer, but no longer applicable to our revised benchmark, as we did not hard-mask or remove any sequences.

> Supplementary fig. 2

Arcadia: Can you make the fonts larger in these figures? This figure is sort of confusing with its separation into panels. I think it would be better to have a single figure with the matching color scheme used for earlier figures in the manuscript (one color per tool), and then draw a darker line for 0.05 error rate, or put an asterisk on the y axis for tools that maintain below that. Same feedback for Fig S3-S7

Response: These figures have been removed in our revision.

> We established the ground truth for these evaluations by computing a full Smith Waterman alignment of the evaluated centroid or cluster member sequences against all centroid sequences using DIAMOND in --swipe mode which guarantees perfect pairwise alignment sensitivity.

Arcadia: Do you think this produces a gold standard ground truth, or will there still be error here? I think highlight possible sources of error here could be helpful for the reader to understand evaluation limitations

Response: No method is perfect and there were sources of error here, however this method of ground truth annotation is no longer used in our revised benchmark. We now use Pfam annotations that are generated by HMMER, which is among the most sensitive known methods to infer sequence homology.

> 22,788,215,153

Arcadia: Can you estimate what fraction of these are overlapping and 100% redundant?

Response: This corresponds to 19,387,935,704 non-redundant sequences. How many of these sequences overlap is vague and cannot be easily answered.

Response to Reviewers

We would like to thank all three reviewers for taking the time to re-assess our significantly revised manuscript and for their efforts in re-reviewing the new version of DeepClust. We agree with all their remaining comments and addressed them in full. Their valuable comments made our benchmarks and the overall manuscript stronger and we hope that DeepClust will become a useful resource for the community.

We had to correct our estimate of novel protein clusters not found in the Big Fantastic database from 93 million to 118 million. This discrepancy is likely caused by sampling bias, as the previous estimate was based on a sample of 10,000 clusters and the current one on 1,000,000 clusters.

We would also like to point out that we carefully shortened the manuscript to meet the 1600 word limit required of a Brief Communication article. Deletions should not be construed as these statements being invalid.

Reviewers' Comments:

Reviewer #1 (Remarks to the Author):

The revised manuscript represents a clear improvement over the previous version. The authors benchmark the most relevant tools including FLSHclust, a recently published method in Science in 2023. The method is technically sound and presents a major improvement in making clustering of large protein sets much more accessible, which are becoming increasingly common (SPIRE, IMG/M, ...). DeepCLUST's scalability across multiple-nodes will allow large scale clustering.

However, the benchmarking and evaluation sections require clarification and revision.

Major points

- The benchmark has improved substantially. In this version, the authors annotated protein sequences with InterPro and generated PFAM architecture strings and use this as ground truth. This is a good benchmark, however, there are still methodological inconsistencies that need to be addressed.

Precision: In the current form it is not clear how the precision is computed, currently it reads "We define the sequence-level precision as the number of sequences in the same cluster of the same domain architecture, divided by the size of the cluster." **How was**

this counted, was it checked if the representative is consistent with the members or was a majority architecture selected to compute the precision? Please clarify this benchmark.

Response: To clarify, the precision is not a number determined per cluster, but assigned to each annotated input sequence individually. For each sequence, precision is calculated as the number of sequences with the same domain architecture within its cluster, divided by the size of the cluster (total number of member sequences). The information whether a sequence is the cluster representative is of no consequence. The distribution over these precision values of individual sequences is then visualized in Fig. 1c, and their arithmetic mean is used as a compounded value. Note that we changed Fig.1c not to overlay violin plots for the precisions because that did not really show any more information and gave the false impression that box plot whiskers extended down to 0.

Sensitivity:

Inconsistent Pfam clan treatment: Precision correctly treats different Pfam families within the same clan as equivalent; however, for sensitivity this is not the case. **Please compute this consistently between the two metrics.**

Response: We now show the sensitivity benchmark evaluation also considering different families in the same Pfam clan as equivalent. As expected, this reduces sensitivity numbers across the board, but the performance of the tools relative to each other does not change. Pfam clans are detected using an elaborate pipeline of profile-profile, structural and functional comparisons (Finn et al., 2006). One would not expect methods based on only pairwise alignment to perform very well at detecting such remote homologs. We have decided to show both evaluations of the benchmark so the reader can consult whichever is more relevant to their application.

Cluster count as recall: An additional recall (sensitivity) independent for the PFAM annotation would be the total number of clusters. The best clustering method should maintain high precision while minimizing the number of clusters. **Please include total cluster counts in Figure 1.**

Response: This has now been added as Fig. 1e.

Sensitivity ceiling: The current benchmark makes achieving 100% sensitivity impossible due to the fixed sequence-identity threshold. Sequences with identical domain architectures but below this identity cutoff of 30% will be cluster separately. **This could be discussed or the definition could be adjusted.**

Response: No identity cutoff was used for the main benchmark. Could the reviewer have gotten this impression because a 30% identity cutoff was used for the big ~22bn experimental study run? This 30% threshold for the experimental study was chosen because the Big Fantastic Database was created using this cutoff, and we wanted to produce a comparable result (in terms of cluster count). We still prefer not to use an identity cutoff for the main benchmark precisely to maximize the sensitivity achievable on pairwise alignment level, and because precision remained high.

- Figure 2 and “dark” proteins: The current “dark” annotation applies a global 90% query-coverage filter, but most functional annotations result from local alignments that do not meet this threshold. Removing the coverage filter would better reveal the truly dark protein fraction. Additionally, relying solely on UMAP visualization can be misleading, as UMAP often distorts global structure and can produce spurious clusters (e.g., 10.1038/s42003-022-03628-x). To increase interpretability, please consider moving the UMAP to the supplement, or fully replacing it with explicit numerical annotations (counts or percentages) for each class.

Response: Some coverage threshold is necessary to sensibly characterize novelty of a sequence. We operate under the assumption that two sequences aligning over 10% of their ranges should not be classified as homologous, as that may just be some promiscuous domain in otherwise different proteins. We now show the fraction of clusters that can be annotated depending on the query coverage threshold, similar to our previous extended data fig. 4, so that users can choose for themselves what may be a good threshold for their application or characterisation of a “dark” proteome. To account for fracturing local alignments, we accumulate query coverage over all HSPs found between a query/target pair. For the domain-level databases, we accumulate query coverage over all hits to the target database. No sequence identity cutoff was used and annotation against the BFD was done using HHblits, as previously requested.

- Discussion regarding Greedy vertex cover. While greedy vertex-based clustering can be useful to generate remote clusterings it comes with the risk of being overly sensitive due to wrong edges in the graph. Additionally, it can easily combine groups that should not be merged. **It would be great to have this as a strength and limitation discussed.**

Response: We have added this discussion in the Greedy vertex cover section to highlight these important strengths and limitations.

- Parameter optimization, DeepClust introduces many parameter choices but does not provide sufficient justification for the decisions. E.g. "default uses a search depth of one for the first two rounds of cascaded clustering, and a depth of three for the remaining rounds.", "In addition, we found it advantageous to use a more stringent clustering criterion in earlier rounds of cascaded clustering and relax it gradually in later rounds.". **Please explain the rationale behind these settings—were they empirically determined, based on theoretical considerations, or borrowed from previous work?**

Response: The parameters were chosen based on the most effective utilization of DIAMOND DeepClust for our main benchmark. To remain fair to the other tools, we also made an effort to find their best possible parameters for the given benchmark (see supplementary information). Minor tweaks like using different cutoffs for different clustering rounds only marginally improved the performance over a more standard baseline. Users can change such parameters on the command line, and their default values were chosen based on standard user expectations.

Minor points

- The manuscript would further improve by including a schematic representation or algorithm that clearly highlights the novelties. While the revisions improved the understandability of the manuscript, some details are still not entirely clear. The manuscript in its current form has enough space to accommodate this addition, which would greatly benefit the reader. I also think that Figures 2 could easily be merged into a single Figure 1, which currently has sufficient available space. Please consider adding this; however, I would leave it to the authors to decide whether or not to implement this suggestion.

Response: We opted not to produce the figure due to the difficulties of adequately visualizing the algorithm within the limited space offered by the print journal. We are space-constrained as our manuscript was at nearly twice the word count permitted by the journal for Brief Communications.

Reviewer #1 (Remarks on code availability):

Code is open source and the method is well maintained.

Response: Thank you!

Reviewer #2 (Remarks to the Author):

The authors have spent tremendous efforts revising the manuscript, with more convincing results added to the revision. They have addressed all my comments.

Reviewer #2 (Remarks on code availability):

I did not try to reproduce the results of the paper, as they are based on very large collection of proteins so it would be difficult even trying to reproduce.

For the github repo, I think the documentation pages can be revised to more clearly show when to use deepclust and how it is different from the other clustering methods ('clust' and 'linclust'). On the wiki page on clustering (<https://github.com/bbuchfink/diamond/wiki/Clustering>), 'deepclust' is hidden deep, and 'diamond deepclust' is not even listed along with 'diamond clust' and 'diamond linclust' among the basic commands.

Response: Thank you for pointing this out to us. We have updated the online Wiki accordingly.

Reviewer #3 (Remarks to the Author):

The manuscript by Buchfink et al. introduces DIAMOND DeepClust, a method for clustering protein sequences at tree-of-life scale with both high sensitivity and computational efficiency. The core contribution is twofold: (1) a cascaded clustering algorithm that leverages the speed of DIAMOND v2 while preserving sensitivity at low sequence identity thresholds, and (2) the application of this method to generate what is, to my knowledge, the largest and most diverse clustered protein dataset to date, comprising 19 billion sequences drawn from 15 public sources.

When I reviewed the earlier version of this manuscript two years ago, I expressed two major reservations: first, the conceptual novelty of the method over existing tools such as Linclust was not clearly articulated, and second, that the benchmarking was insufficiently rigorous and poorly presented, making it difficult to assess whether DeepClust offered a substantive advance. Nevertheless, I noted that the tool showed potential given its promising performance in our own independent testing.

I am pleased to report that the current revision fully addresses the points raised in my previous report. In particular, the benchmarking has significantly improved: it is now clearer, broader in scope, and better controlled. It includes systematic comparisons

across sequence identity regimes and against the most relevant alternatives—MMseqs2-Linclud, MMseqs2-Cluster, and FLSHclust. Most notably, the authors now demonstrate that DeepClust outperforms even MMseqs2-Cluster, which is widely considered a state-of-the-art method, particularly in the low-identity regime that is most relevant for comparative genomics and functional inference across large evolutionary distances.

With the methodological presentation and benchmarking now robust and compelling, I believe this revised manuscript makes a strong case for publication. I just have a couple of minor/discretionary comments below that should be easy to address.

Minor comments

=====

* The authors have removed the results for 90% identity, where DIAMOND DeepClust performs worse in terms of sensitivity. These results could still be reported (e.g. in supplementary) along with a discussion in the main text. The reasoning was that high identity is "not the focus of this work", but a typical user should be aware of this limitation and not assume the package performs best in all scenarios.

Response: In the first version of our manuscript (still available on bioRxiv for reference purposes), we reported runtimes of 0.93h for the linear mode of DIAMOND DeepClust and 7.13h for MMseqs2/Linclud when clustering at 90% identity (Fig. 1). We reported 6.72h for DIAMOND DeepClust (non-linear mode) and 19.4h for MMseqs2. We reported sensitivity errors of 0.121 for the linear mode of DIAMOND DeepClust and 0.15 for MMseqs2/Linclud (lower is better) (Supplementary fig. 2). For the same metric, we reported 0.016 for DIAMOND DeepClust (non-linear mode) and 0.02 for MMseqs2. The figures are admittedly hard to read. This showed that our software also clearly outperforms MMseqs2 and MMseqs2/Linclud when clustering at 90% identity.

* One discretionary suggestion would be to combine Figures 1a and 1b and use a logarithmic scale for CPU hours, although the current format is OK.

Response: We would prefer to keep them as two separate figures to be able to communicate more effectively how the runtimes of the tools differ.

* Another discretionary suggestion would be to include a figure that describes the method, as previously suggested by Reviewer 1.

Response: We opted not to produce the figure due to the difficulties of adequately visualizing the algorithm within the limited space offered by the print journal.

Reviewer #3 (Remarks on code availability):

Code is well documented and could be run without any issue.

Response: Thank you!

Response to Reviewers

Reviewer #1:

Remarks to the Author:

The authors have addressed all my comments.

Response: We are thrilled to learn that we could address all remaining reviewer comments and thank Reviewer #1 for taking the time to re-review our revised manuscript.